# Endothelial alpha globin is a nitrite reductase

T. C. Stevenson Keller IV[1,2], Christophe Lechauve [3], Alexander S. Keller[1,4], Gilson Brás Broseghini-Filho[1,5], Joshua T. Butcher[6], Henry R. Askew Page[1], Aditi Islam[1], Zhe Yin Tan [1], Leon J. DeLalio[1,4], Steven Brooks[7], Poonam Sharma[8], Kwangseok Hong [9], Wenhao Xu[10], Alessandra Simão Padilha [5], Claire A. Ruddiman[1,4], Angela K. Best[1], Edgar Macal[1], Daniel B. Kim-Shapiro[11], George Christ [8], Zhen Yan[1], Miriam M. Cortese-Krott [12], Karina Ricart [13], Rakesh Patel[13], Timothy P. Bender [14], Swapnil K. Sonkusare [1,2], Mitchell J. Weiss [3], Hans Ackerman [7], Linda Columbus[2,15] & Brant E. Isakson [1,2] ✉

Resistance artery vasodilation in response to hypoxia is essential for matching tissue oxygen and demand. In hypoxia, erythrocytic hemoglobin tetramers produce nitric oxide through nitrite reduction. We hypothesized that the alpha subunit of hemoglobin expressed in endothelium also facilitates nitrite reduction proximal to smooth muscle. Here, we create two mouse strains to test this: an endothelial-specific alpha globin knockout (EC Hba1Δ/Δ) and another with an alpha globin allele mutated to prevent alpha globin's inhibitory interaction with endothelial nitric oxide synthase (Hba1WT/Δ36−39). The EC Hba1Δ/Δ mice had significantly decreased exercise capacity and intracellular nitrite consumption in hypoxic conditions, an effect absent in Hba1WT/Δ36−39 mice. Hypoxia-induced vasodilation is significantly decreased in arteries from EC Hba1Δ/Δ, but not Hba1WT/Δ36−39 mice. Hypoxia also does not lower blood pressure in EC Hba1Δ/Δ mice. We conclude the presence of alpha globin in resistance artery endothelium acts as a nitrite reductase providing local nitric oxide in response to hypoxia.

Matching blood perfusion to metabolic demand is a critical function of the autoregulation of the resistance vasculature. Although several vasodilatory signaling pathways have been identified in small arteries, nitric oxide (NO) signaling is among the most potent mechanisms. This is especially relevant in hypoxia, in which NO has been implicated as a key regulator of acute hypoxic vasodilatory responses[1–3].

A major source of hypoxic NO generation is the reaction of the nitrite anion ($NO_2^-$) with metal centers, especially the iron in ery-

[1]Robert M Berne Cardiovascular Research Center, University of Virginia School of Medicine, Charlottesville, VA, USA. [2]Department of Molecular Physiology and Biophysics, University of Virginia School of Medicine, Charlottesville, VA, USA. [3]Department of Hematology, St. Jude Children's Research Hospital, Memphis, TN, USA. [4]Department of Pharmacology, University of Virginia School of Medicine, Charlottesville, VA, USA. [5]Department of Physiological Sciences, Federal University of Espirito Santo, Vitória, Brazil. [6]Department of Physiological Sciences, College of Veterinary Medicine, Oklahoma State University, Stillwater, OK, USA. [7]Physiology Unit, Laboratory of Malaria and Vector Research, National Institute of Allergy and Infectious Diseases, Bethesda, MD, USA. [8]Department of Biomedical Engineering, University of Virginia, Charlottesville, VA, USA. [9]Department of Physical Education, College of Education, Chung-Ang University, Seoul, South Korea. [10]Transgenic Mouse Facility, Department of Medicine, University of Virginia School of Medicine, Charlottesville, VA, USA. [11]Department of Physics, Translational Science Center, Wake Forest University, Winston-Salem, NC, USA. [12]Cardiovascular Research Laboratory, Division of Cardiology, Pneumology and Angiology, Medical Faculty, Heinrich Heine University, Düsseldorf, Germany. [13]Department of Pathology, University of Alabama at Birmingham, Birmingham, AL, USA. [14]Department of Microbiology, Immunology and Cancer, University of Virginia School of Medicine, Charlottesville, VA, USA. [15]Department of Chemistry, University of Virginia, Charlottesville, VA, USA. ✉e-mail: brant@virginia.edu

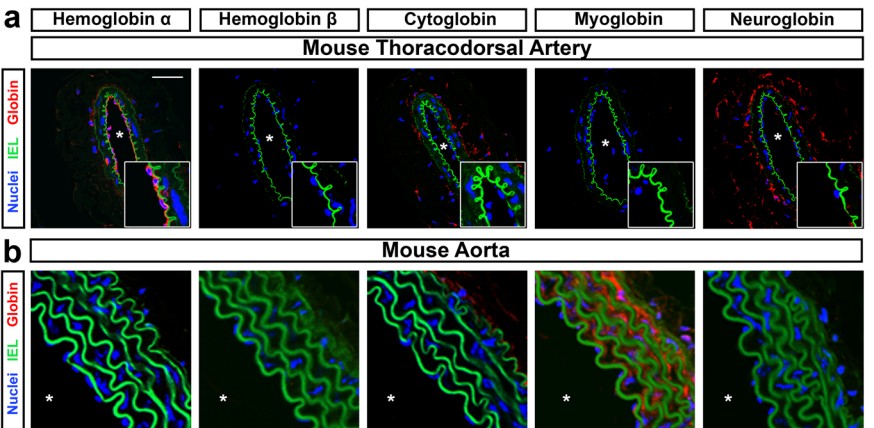

**Fig. 1 | Alpha globin is the only globin expressed in the mouse endothelium.**
**a** The mouse thoracodorsal artery (a skeletal muscle-derived resistance artery) shows endothelial expression of alpha globin, but not of other globins. Cytoglobin and neuroglobin are observed in adventitial layers, but not in endothelial cells. The scale bar represents 50 µm; magnified views focused on the endothelium are shown in the bottom right of each image. **b** The mouse aorta is representative of globin expression in a conduit artery. Only myoglobin is expressed in the aorta wall, and it is not expressed in endothelium. In all images, green signal is autofluorescence of the internal elastic lamina layer (IEL) and the asterisk indicates the lumen of the vessel. All images are representative of observations from a minimum of four mice per staining condition.

throcytic hemoglobin[2,4]. This reaction occurs with deoxygenated heme groups, with nitrite being reduced to vasoactive NO. Many studies from independent laboratories have implicated erythrocyte NO generation and/or delivery as a critical source of hypoxic vasodilation, although questions remain about the nature and stability of the molecule carrying "NO bioactivity"[5–8]. Other sources of NO for hypoxia-driven vasodilation in the resistance vasculature have yet to be determined.

Our group has previously described alpha globin in the endothelium of resistance arteries localized specifically at the site of endothelial and smooth muscle interaction, i.e., the myoendothelial junction (MEJ)[9–11]. Here, alpha globin acts as a negative regulator of endothelial NO synthase (eNOS)-derived signaling through NO scavenging aided by direct interaction with eNOS[9,10,12,13]. Negative regulation of NO signaling by endothelial alpha globin is $O_2$ dependent, as both eNOS production of NO and deactivation of NO by alpha globin (producing nitrate, $NO_3^-$) require $O_2$. Recent human studies have confirmed a role for endothelial alpha globin in NO catabolism, as alpha thalassemic individuals demonstrated increased flow-mediated dilation[14].

Endothelial alpha globin is also a candidate for controlling hypoxia-induced vasodilation by generating NO through nitrite reduction. Alpha globin expression in the MEJ provides a pool of hemoproteins that are optimally positioned next to vascular smooth muscle to control vasodilation. Endothelial alpha globin might act as a sensor and actuator of hypoxia-driven signaling, effecting precise control of local vasodilation to perfuse metabolically active tissues during exercise or other acute hypoxia scenarios. We hypothesized that endothelial alpha globin had a critical secondary role in controlling vasodilation, separate from its catabolic role in vascular tone homeostasis by interaction with eNOS and sequestration of NO. These differential actions of endothelial alpha globin are dependent on local oxygen tension and tissue oxygen demand: when oxygen is not critically needed by the tissue, endothelial alpha globin might act primarily as a block to NO signaling from the endothelium and prevent vasodilation; however, when tissue demand for oxygen is high, endothelial alpha globin might switch roles to promote nitrite reduction and vasodilation to increase tissue perfusion.

To test whether endothelial alpha globin contributed to hypoxic vasodilation signaling, we generated two genetic mouse models: one with endothelial-specific loss of alpha globin through *Cre/lox* recombination and a CRISPR-based mutant in which alpha globin is present but lacks its eNOS-binding domain. Using these two models, we have decoupled the function of alpha globin as an eNOS-interacting NO scavenger from its role as a nitrite reductase. We have also found deficiencies in hypoxia-induced vasodilation, as well as in global physiologic measurements of exercise capacity and blood pressure regulation, but only when the full alpha globin protein is deleted from the endothelium. We demonstrate that endothelial alpha globin acts as a nitrite reductase in the vessel wall, thereby further refining the roles of alpha globin in regulating vascular NO homeostasis.

## Results

### Alpha globin is the only globin expressed in the endothelium of mouse resistance vasculature

Expression of the alpha subunit of hemoglobin has been observed in the endothelium of small resistance arteries by our group and others[10–12,15]. Before targeting alpha globin, we assayed the expression of other mammalian globins (including myoglobin, cytoglobin, and neuroglobin) that could catalyze nitrite reduction in the vascular wall of resistance arteries. Using the murine skeletal muscle-supplying thoracodorsal artery (which has an average diameter of <200 µm) as a model of resistance arteries, and the murine aorta as a representative conduit artery, we examined the expression of globin isoforms in endothelium and smooth muscle layers with confocal microscopy (Fig. 1). Human small arteries from adipose biopsies were also used for a translational comparison (Supplementary Fig. 1). In the murine thoracodorsal artery, a strong signal for alpha globin was observed in the endothelium (Fig. 1a, vessel lumen is denoted by an asterisk). No other globin isoforms were observed in the endothelium of the mouse thoracodorsal artery, though cytoglobin and neuroglobin appeared to localize to adventitial layers (Fig. 1a). Similarly, alpha globin was the primary endothelial globin observed in our cohort of tested human microvessels. However, other studies have observed the expression of hemoglobin β mRNA in human endothelium, indicating it may also be present at different expression levels or vascular beds[12]. We observed diffuse staining for cytoglobin, myoglobin, and neuroglobin in the smooth muscle and adventitial layers of the human vessels (Supplementary Fig. 1a). Alpha globin expression in the endothelium is characteristic of resistance vessels and is not found in all vessel types; in the aorta, alpha globin was absent from the endothelium (Fig. 1b). In contrast, a strong myoglobin signal was observed in the multiple smooth muscle layers of the aorta (Fig. 1b). Antibody isotype controls were analyzed (Supplementary Fig. 1b), and a positive

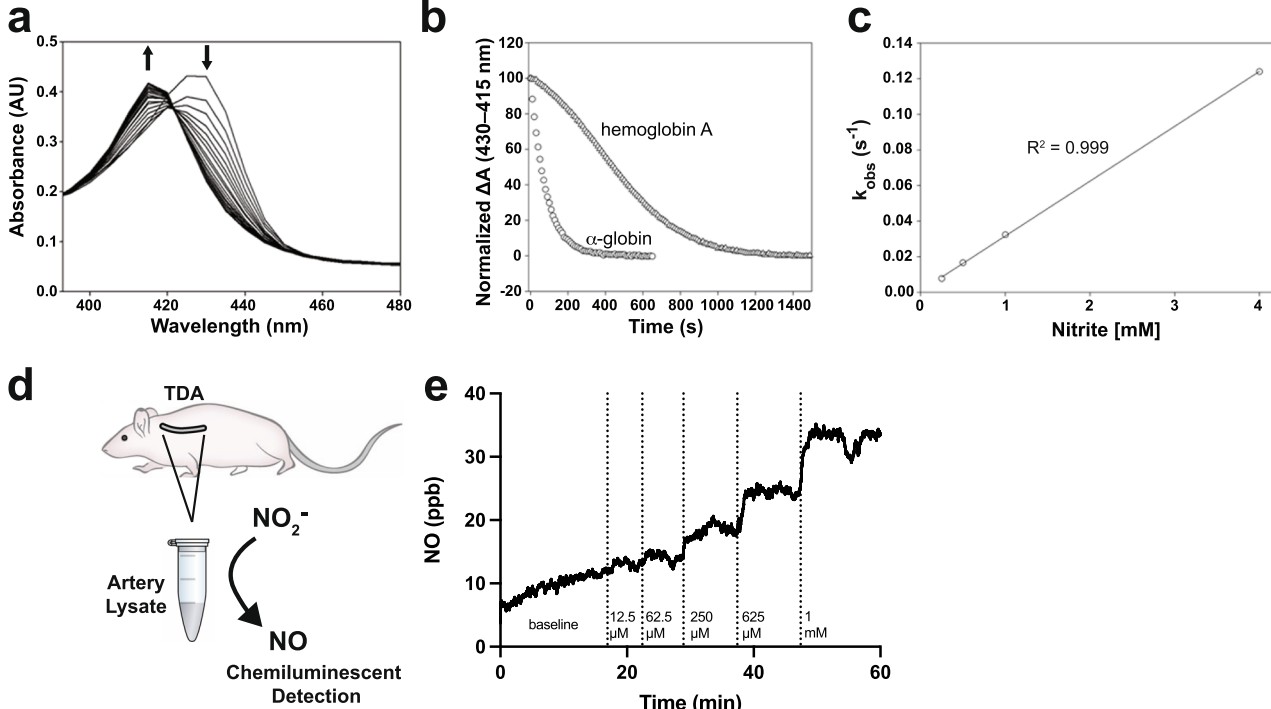

**Fig. 2 | Alpha globin is a nitrite reductase. a** Nitrite-induced optical changes to deoxyferrous alpha globin caused by NO gas formation and binding are reflected in the changing spectra over time. Sodium nitrite solution (1 mM) was added to deoxyferrous alpha globin (4 μM), and spectra were recorded at 10-s intervals. **b** Example kinetic traces of ferrous-NO formation, as measured by the change in absorbance at 415 nm and 430 nm as a result of the reaction of alpha globin (circles) or hemoglobin A (triangles) and nitrite. **c** The observed rate constant for the formation of ferrous-NO from the reaction of (deoxy) alpha globin and nitrite across nitrite concentrations. **d** Illustration of the experimental design for observing NO generation from nitrite added to thoracodorsal artery lysate in a chemiluminescent detection setup. **e** Representative plot of NO generation from thoracodorsal artery lysate with increasing doses of nitrite added (the graph is representative of $n = 4$ experiments). Vertical lines represent time points at which the indicated concentrations of nitrite were added to the solution to observe NO formation (measured in ppb). Source data are provided in the Source data file.

immunofluorescence signal for hemoglobin β was observed in erythrocytes (Supplementary Fig. 2).

## Alpha globin has nitrite reductase function

As alpha globin is the only globin isoform in the thoracodorsal artery, we tested whether alpha globin could generate NO via nitrite reduction in vitro and in isolated vessels ex vivo. First, isolated deoxyferrous alpha globin was reacted with nitrite in the presence of sodium dithionite ($Na_2S_2O_4$) to form NO-bound ferrous alpha globin (Fig. 2a–c). Importantly, nitrite does not react with $Na_2S_2O_4$ in vitro[16]. We showed that 1 mM $Na_2S_2O_4$ induced lower solution oxygen tension to mimic hypoxic environments (Supplementary Fig. 3). Optical changes show a hypsochromic shift of the Soret peak from 430 to 415 nm with increasing reaction time. The biomolecular rate constant of nitrite reduction by deoxyferrous alpha globin is $35 \pm 3 \, M^{-1} s^{-1}$ at 25 °C and pH 7.4 ($R^2 = 0.999$ for the exponential fit) (Fig. 2b). The observed rate constant for nitrite reduction by isolated alpha globin as a function of nitrite concentration shows linear dependence (Fig. 2c). Nitrite reduction by the hemoglobin α2β2 tetramer (HbA) showed more complex kinetics, with HbA changing from deoxyferrous HbA to ferrous nitrosyl HbA (Fig. 2b). The rate constant we observe for the hemoglobin tetramer is in agreement with previous analyses at approximately $12 \, M^{-1} s^{-1}$ [17]. Furthermore, we tested the ability of endothelial alpha globin to reduce nitrite and form NO from isolated vessels. Pooled lysates of thoracodorsal arteries from WT mice (as used above) were subjected to chemiluminescent detection of NO with increasing doses of nitrite (Fig. 2d). NO generation from thoracodorsal artery lysate increased with increasing doses of nitrite, demonstrating

NO generation in arteries that strongly express endothelial alpha globin (Fig. 2e).

## Generation of an Hba1 conditional allele

To test the role of alpha globin in the endothelium, we generated mice with a mutant *Hba1* allele with *loxP* recombination sites flanking exons 2 and 3 (Fig. 3a). These mutant mice, bred to homozygosity (*Hba1^fl/fl*), had with no observed lethality (Supplementary Table 1). Temporal and endothelial cell-type-specific knockout of *Hba1* was driven by the tamoxifen-inducible Cre recombinase downstream of a VE-cadherin promoter (*Cdh5-PAC-Cre^ERT2*)[18] (denoted as EC *Hba1^Δ/Δ* hereafter) (Fig. 3a). After recombination, PCR amplification of the *Hba1* locus demonstrates a smaller amplification product in EC *Hba1^Δ/Δ* mice (450 bp, as compared to 1100 bp in control animals) as a consequence of the deletion of exons 2 and 3 (Fig. 3b) with no change in the expected birth ratios with the Cre driver (Supplementary Table 2). Endothelial-specific knockout of *Hba1* reduced the immunofluorescent signal of alpha globin in the endothelium (Fig. 3c–e), which was similar to the expression in an *Hba1* global-knockout model (Supplementary Fig. 4). Specifically, alpha globin was localized to holes in the IEL where myoendothelial junctions form, as shown in *en face* immunofluorescence confocal images of the thoracodorsal artery (Fig. 3c–e and Supplementary Fig. 5). Clinically, this model is most similar to a trans-two alpha gene deletion, in which only one of the two copies of the alpha globin gene is lost. However, this genetic deletion only affects alpha globin expression in the endothelium. Endothelial-specific deletion was confirmed, with no difference in RBC counts in EC *Hba1^Δ/Δ* mice, as would be expected from the loss of *Hba1* in cells of

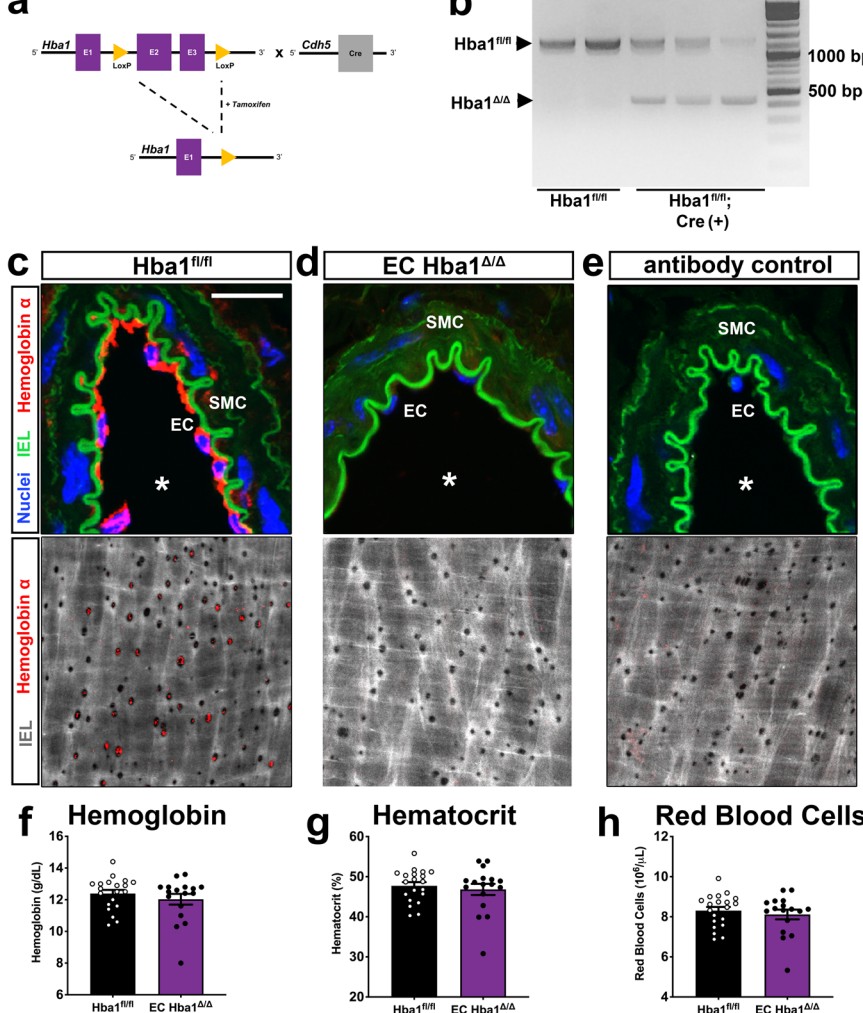

**Fig. 3 | Creation of an endothelial-specific Hba1-deletion mouse model.**
**a** Deletion strategy: *loxP* sites flanking exons 2 and 3 of the mouse *Hba1* gene (*Hba1^fl/fl*) were introduced by recombineering. A tamoxifen-inducible, endothelial-specific Cre recombinase (*Cdh5-PAC-Cre^ERT2*) enabled temporally controlled and cell-type-specific deletion of a functional *Hba1* gene (EC *Hba1^Δ/Δ* mouse). **b** DNA gel using genomic DNA extracted from diaphragm showing recombination of the *Hba1* locus with Cre activation. The recombination event produced a band at ~450 bp. **c**, **d** Immunofluorescence staining for alpha globin in transverse sections of a thoracodorsal artery (top) and en face views of the endothelium of a third-order mesenteric artery (bottom). The scale bar represents 25 μm, and an asterisk indicates lumen of the vessel in the upper images. Alpha globin (red) was found in the endothelium throughout and specifically in the holes in the internal elastic lamina (IEL), where myoendothelial junctions are found. **e** Control IgG staining showing the specificity of the staining for alpha globin in this tissue. **f, g** Endothelial deletion of alpha globin does not affect blood cell hemoglobin parameters. Blood hemoglobin content (**f**), hematocrit (**g**), and the number of red blood cells (**h**) were unchanged in the EC *Hba1^Δ/Δ* mice, as compared to the *Hba1^fl/fl* controls. For the experiments in **f–h**, n = 20 *Hba1^fl/fl* mice and n = 17 EC *Hba1^Δ/Δ* mice were used; one-sided *t* tests were used to determine whether there were significant differences between groups. Bar graphs are centered on mean with error bars denoting standard error. Source data are provided in the Source data file.

hematopoietic lineages (Fig. 3f–h). There were no changes in erythrocyte hemoglobin protein, erythrocyte size, or erythrocyte volume upon *Cdh5-Cre*–driven deletion (Supplementary Fig. 6); this is in contrast to the decreased erythrocyte volume and anemia in *Hba1* global knockout animals (*Hba1^-/-*, Supplementary Figs. 6 and 7). Global recombination of our *Hba1* conditional allele (*Hba1^fl/fl; Sox2-Cre*[19]), hereafter referred to as global *Hba1^Δ/Δ*, (Supplementary Fig. 7a–c) produced an anemia phenotype that mirrors the global loss of *Hba1* through the insertion of a neomycin resistance cassette[20] (Supplementary Fig. 7d–f) with significant defects in blood hemoglobin and hematocrit but with no statistically significant change in total RBCs.

Furthermore, the specificity of the recombination event for the *Hba1* gene locus was demonstrated by the deletion with the *Sox2-Cre* transgenic driver. If both *Hba1* and *Hba2* genes were affected by the *Cre/lox* recombination, severe alpha thalassemia would be induced, resulting in embryonic lethality[21]. Global *Hba1^Δ/Δ* mice are

born at the expected Mendelian ratios (Supplementary Table 3, compare with the genotypes from the global *Hba1^-/-* model in Supplementary Table 4), suggesting that only the *Hba1* locus is flanked by recombination sites, even though the *Hba1* and *Hba2* loci share high sequence similarity.

### Generating a genetic model to disrupt alpha globin/eNOS interaction
Endothelial-specific knockdown of the total alpha globin protein prevented interrogation of our hypothesized nitrite reduction role independent of its known role in NO scavenging through association with eNOS[9,10]. To uncouple the effects of loss of the heme site chemistry and eNOS binding, we generated a murine model in which alpha globin specifically lacks the identified eNOS-interacting sequence (*Hba1* residues 34–43)[9,10]. By using CRISPR/Cas9 gene editing to target the region of alpha globin that interacts with eNOS (Fig. 4a), we generated

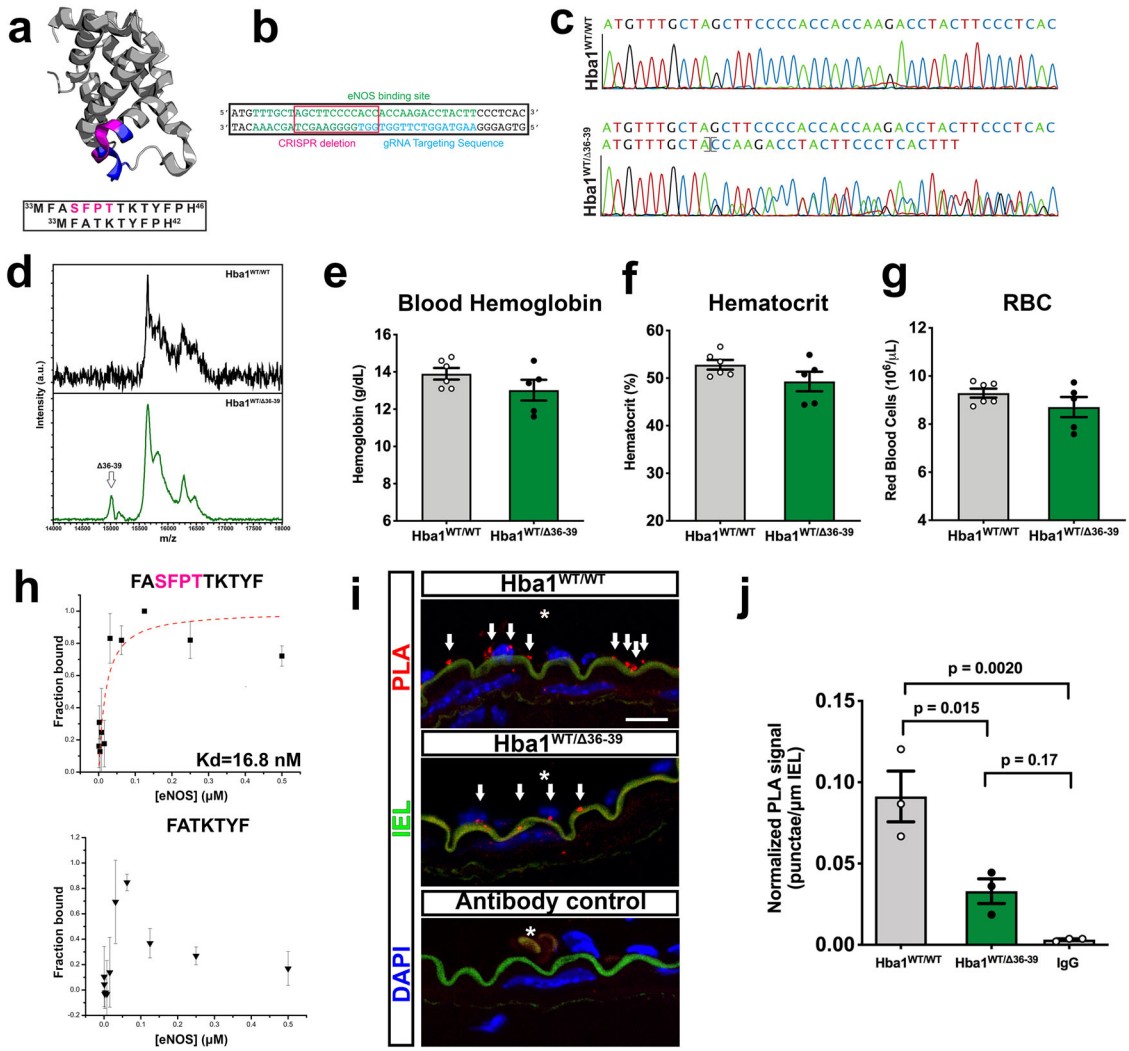

**Fig. 4 | Disruption of alpha globin/eNOS interaction through the deletion of four residues in Hba1. a** A colorized crystallographic model of alpha globin (from PDBid: 1Z8U) showing the residues previously shown to interact with eNOS (blue) surrounding four residues deleted in the *Hba1^{WT/Δ36−39}* mouse model (magenta). The sequences for the proteins encoded by the *Hba1^{WT}* and *Hba1^{Δ36−39}* alleles, including the deleted residues (magenta), are shown in the box below. **b** Using a guide RNA design (blue) targeting the eNOS binding region (green) resulted in an in-frame deletion of the nucleotides boxed in pink. This is confirmed by the chromatogram in **c**, which shows that the 12-nucleotide deletion scrambles downstream reading of the nucleotide sequence in NGS protocols. **d** The mutant protein is expressed, as seen in immunoprecipitation-coupled mass spectrometry. A peak corresponding to a protein with a molecular weight ~400 Da less than that of the dominant species is seen in hemoglobin captured from lysed red blood cells. **e–g** The blood hemoglobin content (**e**), hematocrit (**f**), and the number of red blood cells (**g**) are shown for the *Hba1^{WT/WT}* and *Hba1^{WT/Δ36−39}* groups. **h** Results of fluorescence polarization assays for determining the binding of the mutant allele to eNOS, using an alpha globin mimetic peptide known to bind (top) and the Δ36−39 peptide (bottom). No binding affinity

could be calculated for the Δ36−39 peptide. The points are centered on the mean value of three measurements per concentration, and the error bars represent the standard deviation of the triplicate measurements. These results are representative of $n > 5$ individual experiments. **i** The proximity ligation assay (PLA) signal (red puncta, marked by arrows) highlights the close localization of alpha globin and eNOS. Nuclei appear blue (with DAPI staining), the autofluorescence of the internal elastic lamina appears green, and the lumen of the vessel is indicated by an asterisk (*). The scale bar represents 15 μm. **j** Endothelial PLA signals, normalized to the length of the internal elastic lamina, were reduced in the *Hba1^{WT/Δ36−39}* mice. The difference was significant when these mice were compared to the *Hba1^{WT/WT}* mice, but not when they were compared to IgG staining controls. For the experiments in **e–g**, $n = 6$ for *Hba1^{WT/WT}* mice and $n = 5$ for *Hba1^{WT/Δ36−39}* mice were used. For the experiments in **j**, $n = 3$ mice of each genotype were used, and the significance of differences was determined by a one-sided *t* test with multiple comparisons correction. Bar graphs are centered on mean with error bars denoting standard error. Source data are provided in the Source data file.

a genetic model with a mutation (an in-frame deletion of 12 nucleotides, Fig. 4b, c) that removed *Hba1* residues 36−39 (NH$_2$−SFPT−COOH). We refer to this allele as *Hba1^{Δ36−39}*. The expression of the mutant alpha globin protein was observed in immunoprecipitate from lysed RBCs, in which an approximate 400 Da peak shift (corresponding to the loss of four amino acids from the polypeptide chain) was observed in a matrix-assisted laser desorption ionization mass spectrometer (Fig. 4d).

The global deletion of these four residues was embryonically lethal by E14.5 in homozygosity (*Hba1^{Δ36−39/Δ36−39}*) (Supplementary

Table 5), with the embryos showing growth restriction on gross observation, dilated major vessels, and eccentrically dilated, non-compact myocardium upon histologic examination (Supplementary Fig. 8a, b). A large increase in tyrosine nitration (detected by immunofluorescence staining) was a possible downstream effect of oxidative and nitrosative stress due to high-output NO generation during development (Supplementary Fig. 8c). As homozygous mutants were not obtained in sufficient numbers for experimentation, so we used heterozygous *Hba1^{WT/Δ36−39}* mice which were fertile and viable into adulthood (Supplementary Table 5).

First, we determined whether the $Hba1^{\Delta36-39}$ mutation affected erythrocyte numbers in adult heterozygous mice. $Hba1^{WT/\Delta36-39}$ mice did not present with abnormalities in total blood hemoglobin, hematocrit, or RBC number (Fig. 4e–g). Furthermore, there were no observed differences in mean corpuscular volume or mean corpuscular hemoglobin for RBCs (Supplementary Fig. 9a, b). Erythrocytes from the $Hba1^{WT/\Delta36-39}$ mice did not appear to turn over more rapidly, as the percentage of reticulocytes was unchanged (Supplementary Fig. 9c). Hemoglobin tetramers from the $Hba1^{WT/\Delta36-39}$ mice were stable, as no insoluble alpha globin was found in blood samples isolated from these mice (Supplementary Fig. 9d). Additionally, their RBC morphology was normal, with no indication of increased hemoglobin precipitation within the RBCs stained with cresyl blue (Supplementary Fig. 9e, f).

To determine whether the deleting $^{36}$SFPT[22] in alpha globin caused the loss of alpha globin/eNOS interaction, we tested the binding affinity of the mutant alpha globin sequence for eNOS in vitro by using a fluorescence polarization binding assay (Fig. 4h). A peptide corresponding to the region of alpha globin that binds with eNOS ($Hba1$ residues FASFPTTTKTYF, Fig. 4h, top) bound to the recombinant eNOS oxygenase domain with an affinity similar to that seen in previous studies (16.8 nM)[9]. In contrast, the $\Delta36-39$ deletion peptide (Fig. 4h, bottom) did not bind to the eNOS oxygenase domain with affinity detectable using this methodology. The loss of the interaction was assayed in ex vivo vessels by a proximity ligation assay (PLA). Using this technique, we found the interaction of alpha globin and eNOS to be significantly reduced in the endothelium of the $Hba1^{WT/\Delta36-39}$ mice (Fig. 4i, j). Arteries from $Hba1^{WT/\Delta36-39}$ and $Hba1^{WT/WT}$ mice had no observable difference in alpha globin and eNOS expression in the endothelium (Supplementary Fig. 10).

The $Hba1^{WT/\Delta36-39}$ mice were hypothesized to have increased vascular NO signaling due to decreased alpha globin/eNOS interaction in the endothelium. Consistent with the increased tyrosine nitration observed in the $Hba1^{\Delta36-39/\Delta36-39}$ embryos, adult $Hba1^{WT/\Delta36-39}$ mice had significantly increased NO production after cholinergic stimulation (Supplementary Fig. 11). This finding is similar to previous results in which the exogenous alpha globin mimetic peptide HbαX increased NO signaling[9,10,13]. Therefore, we were able to create a mouse model that mimicked the effects of eNOS and alpha globin loss of interaction (as in prior work with HbαX) but that retained alpha globin expression in the endothelium.

## Treadmill running distance is moderately decreased in EC Hba1$^{\Delta/\Delta}$ mice

A major physiologic role of hypoxic vasodilation is regulating blood perfusion to match metabolic demand. This is particularly relevant in the context of exercise-induced hyperemia in skeletal muscles. A forced-exercise model can recapitulate conditions of relative hypoxia in skeletal muscle[23,24]; therefore, we used such an experimental approach to determine whether the loss of nitrite reductase function by knockout of endothelial alpha globin decreased vasodilation in response to hypoxic stress and, therefore, exercise fitness. Mice were encouraged to run on a treadmill with controlled speed until exhaustion (Fig. 5a). There were no differences in body weight between littermate controls of the EC $Hba1^{\Delta/\Delta}$ and $Hba1^{fl/fl}$ mice or the $Hba1^{WT/WT}$ and $Hba1^{WT/\Delta36-39}$ mice (Fig. 5b). The distance to exhaustion was significantly shorter for the EC $Hba1^{\Delta/\Delta}$ group than for their $Hba1^{fl/fl}$ littermates, whereas $Hba1^{WT/WT}$ and $Hba1^{WT/\Delta36-39}$ littermate pairs were exhausted at a similar distance (Fig. 5c). Blood lactate measurements (a metric for muscular workload) did not differ significantly between littermate-paired groups as measured with arterial/veinous blood from tail snips (Fig. 5d). Similar to the EC $Hba1^{\Delta/\Delta}$ mutation, global knockout of $Hba1$ (in the global $Hba1^{\Delta/\Delta}$ model) resulted in a significant decrease in running distance (Supplementary Fig. 12). The effects of mouse genotype on running distance were independent of differences in

soleus muscle capillary density between littermate controls (Fig. 5e). Taking these results together, it appears that the loss of endothelial alpha globin protein and, therefore nitrite reduction capacity, is detrimental to exercise capacity.

## Endothelial alpha globin controls vessel diameter through nitrite reduction

To determine whether endothelial alpha globin could affect individual arterial responses to hypoxia through nitrite reduction, we measured intracellular nitrite and nitrate concentrations from isolated thoracodorsal arteries. Isolated arteries were incubated with increasing doses of $Na_2S_2O_4$ were incubated with the isolated arteries to recapitulate the hypoxia experienced by skeletal muscle arteries during exercise (Supplementary Fig. 3). In control WT vessels, we found a significant decrease in nitrite concurrent with increased cellular nitrate with $Na_2S_2O_4$ treatment or $N_2$ gas-treated (deoxygenated) buffer, but not in vessels in Krebs buffer treated with $H_2O$ (vehicle) alone (Fig. 6a and b). Importantly, there was no difference in total $NO_2^-$ and $NO_3^-$ species across littermate groups in any of the experiments (Fig. 6c). Cellular $NO_2^-$ was significantly increased after $Na_2S_2O_4$ treatment in both global $Hba1^{-/-}$ and EC $Hba1^{\Delta/\Delta}$ vessels with $Na_2S_2O_4$ treatment (Fig. 6a), as compared to their respective controls, with no change in total $NO_2^-$ and $NO_3^-$ (Fig. 6c). Again, $NO_2^-$ consumption was not recapitulated with $H_2O$ treatment. In the $Hba1^{WT/\Delta36-39}$ mice, nitrite consumption was unchanged compared to that in littermates, and no effect was observed with $H_2O$ treatment (Fig. 6a). These data indicate that the isolated artery itself, independent of erythrocytes, can consume $NO_2^-$ in response to hypoxia, and that $NO_2^-$ consumption correlates with the presence of endothelial alpha globin.

The vasodilatory response of isolated arteries induced by hypoxia was measured using pressure myography (Fig. 7a). In mice with genotypes in which endothelial alpha globin is intact and can participate in hypoxic nitrite reduction (WT, $Hba1^{+/+}$, $Hba1^{fl/fl}$ and both $Hba1^{\Delta36-39}$ genotypes), a robust dilation in response to $Na_2S_2O_4$ was observed. However, in the $Hba1^{-/-}$ and EC $Hba1^{\Delta/\Delta}$ groups, vasodilation was blunted across increasing doses of $Na_2S_2O_4$ (Fig. 7b). The basal tone of the arteries and responses to 1 μM NS309 from littermate groups did not differ (Supplementary Fig. 13). Other experiments with pharmacologic agents were performed with a single dose of 1 mM $Na_2S_2O_4$, as that is in the middle of our range and produced a robust dilation (Fig. 7c).

As a control, we reintroduced RBCs into the lumens of EC $Hba1^{\Delta/\Delta}$ arteries to determine whether erythrocyte-dependent nitrite reduction was sufficient to restore a dilatory response to $Na_2S_2O_4$-induced hypoxia. A statistically significant, yet partial, the rescue of vasodilation was observed with arteries and luminal erythrocytes treated with $Na_2S_2O_4$ (Fig. 7d).

To determine the source of vasodilatory signals, we tested the vessel dilatory response in the presence of various pharmacologic agents targeting NO signaling and enzymatic production of NO. First, to block NO production from NOS isoforms, isolated arteries were incubated with the NOS inhibitor L-nitroarginine methyl ester (L-NAME) before being subjected to $Na_2S_2O_4$-induced hypoxia (Fig. 7e). NOS inhibition did not affect the robust dilation in the $Hba1^{+/+}$, $Hba1^{fl/fl}$, $Hba1^{WT/WT}$, or $Hba1^{WT/\Delta36-39}$ mice. Dilation of the arteries of $Hba1^{-/-}$ and EC $Hba1^{\Delta/\Delta}$ mice remained low in this context; therefore, the hypoxic vasodilation response appears to be NOS-independent but relies on the presence of the full alpha globin protein.

Nitrite reduction is dependent on heme redox chemistry. To prevent nitrite reduction by endothelial alpha globin heme iron, we pre-incubated isolated vessels with carbon monoxide (CO), which tightly binds the heme group and prevents heme-based catalysis and nitrite reduction (Fig. 7f). Treatment with CO fully blocked hypoxic vasodilation responses of all groups to levels equivalent to $Hba1^{-/-}$ and EC $Hba1^{\Delta/\Delta}$ mice; therefore, the hypoxic vasodilation is dependent on

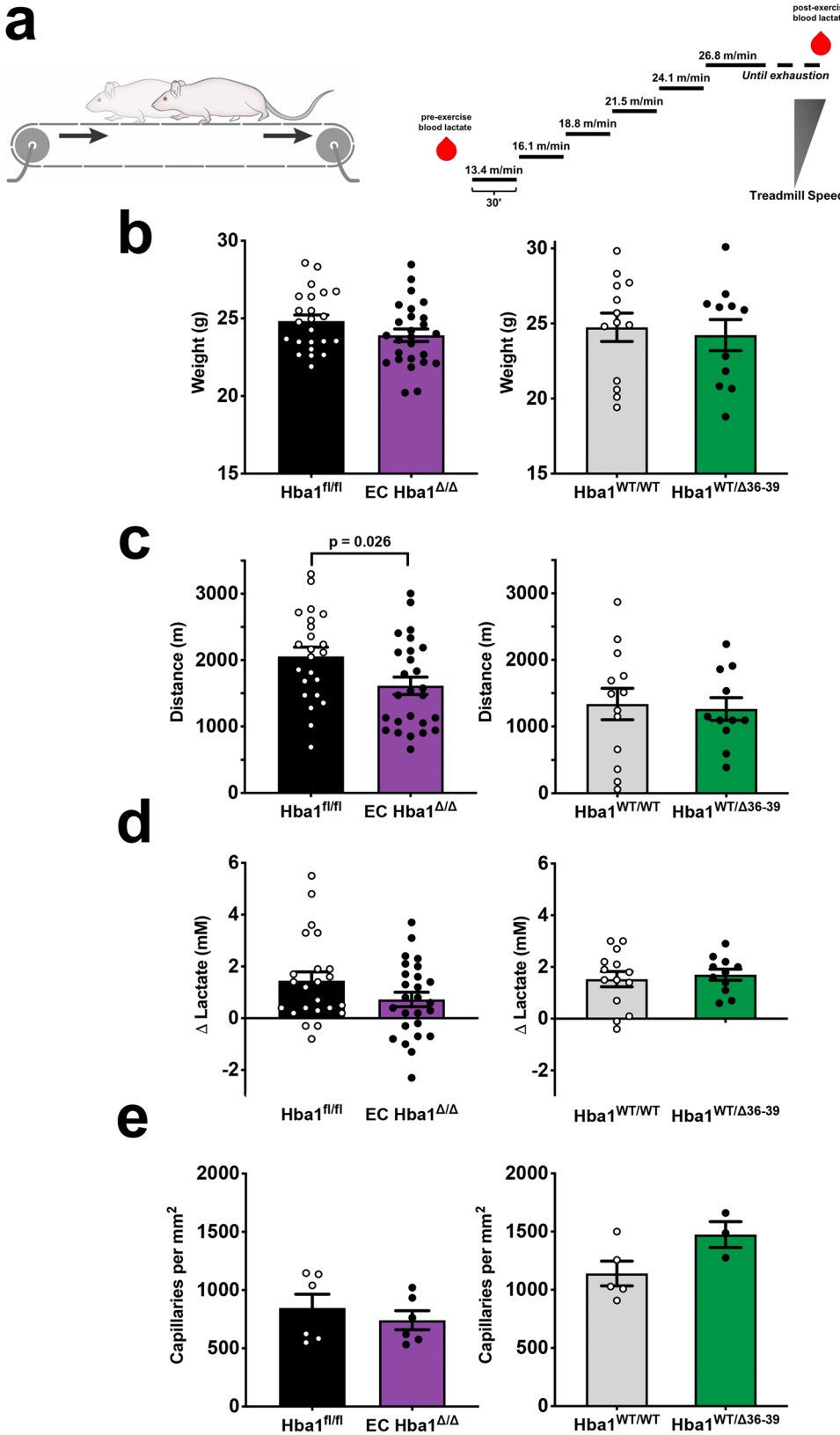

redox chemistry that can be abrogated by CO. Furthermore, incubating the vessel with the NO scavenger PTIO (2-phenyl-4, 4, 5, 5,-tetra-methylimidazoline-1-oxyl 3-oxide) blunted dilation in all groups, demonstrating that the hypoxic vasodilation is dependent on NO signaling (Fig. 7g).

Lastly, other vasoactive molecules produced by endothelium might contribute to the hypoxic vasodilation response. Hydrogen peroxide or other reactive oxygen species could cause dilation in hypoxia; however, treating the isolated vessels with the peroxide scavenger TEMPOL did not affect hypoxia-induced dilation for any

**Fig. 5 | Exercise capacity is reduced with total loss of function, but not with eNOS binding disruption. a** Schematic of the exercise capacity protocol. After an initial blood sample was collected for baseline lactate measurement, mice were encouraged to run on a treadmill until exhaustion. The treadmill speed was increased every 30 min until a maximum speed of 26.8 m/min (1 mile per hour) was achieved. After running failure, a second blood sample was taken to monitor lactate buildup and thereby ensure physical exhaustion. **b** Body weight measurements for $Hba1^{fl/fl}$ vs. EC $Hba1^{\Delta/\Delta}$ or for the $Hba1^{WT/WT}$ vs. $Hba1^{WT/\Delta36-39}$ littermate groups did not differ. **c** The distance to exhaustion was shorter in EC $Hba1^{\Delta/\Delta}$ mice compared to $Hba1^{fl/fl}$ littermates; no such difference was observed when the distances to

exhaustion for the $Hba1^{WT/WT}$ and $Hba1^{WT/\Delta36-39}$ littermate groups were compared. **d** Blood lactate was increased in all groups after exercise, but no differences were observed between the control and experimental groups. **e** Soleus muscle capillary density was similar across littermate comparisons. For the experiments in **b–d**, $n = 23$ for $Hba1^{fl/fl}$ mice; $n = 26$ for EC $Hba1^{\Delta/\Delta}$ mice; $n = 13$ for $Hba1^{WT/WT}$ mice; and $n = 11$ $Hba1^{WT/\Delta36-39}$ mice were used. For the experiments in **e**, $n = 6$ for $Hba1^{fl/fl}$ mice; $n = 6$ for EC $Hba1^{\Delta/\Delta}$ mice; $n = 5$ for $Hba1^{WT/WT}$ mice; and $n = 3$ for $Hba1^{WT/\Delta36-39}$ mice were used. Results for different littermate genotypes were compared with $t$ tests. Bar graphs are centered on mean with error bars denoting standard error. Source data are provided in the Source data file.

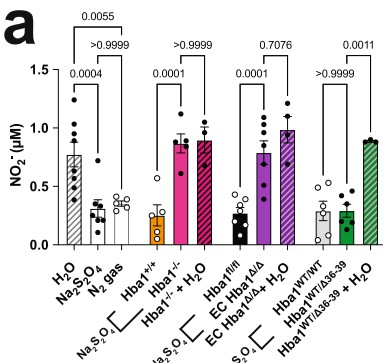 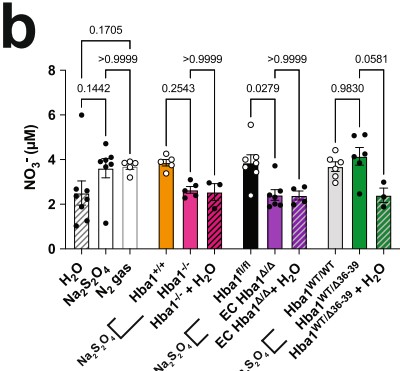 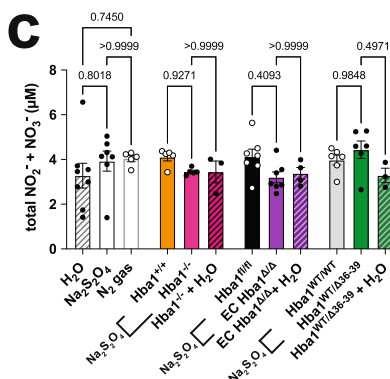

**Fig. 6 | Nitrite consumption is increased by chemical hypoxia in isolated thoracodorsal vessels.** Nitrite ($NO_2^-$) (**a**), nitrate, ($NO_3^-$) (**b**), and summed $NO_2^-$ and $NO_3^-$ species (**c**) were measured after isolated vessels were incubated with sodium dithionite ($Na_2S_2O_4$). Wild-type thoracodorsal vessels treated with $Na_2S_2O_4$ or buffer deoxygenated with $N_2$ gas showed decreased intracellular $NO_2^-$ when compared to vessels treated with water ($H_2O$) (leftmost group). All experimental genotypes and their littermate controls were treated with $Na_2S_2O_4$, and the experimental genotypes were treated with $H_2O$ as a vehicle control (cross-hatched bars). Vessels from both global Hba1$^{-/-}$ and EC $Hba1^{\Delta/\Delta}$ mice showed higher levels of

$NO_2^-$ and lower $NO_3^-$ levels when compared to those of littermate controls with chemical hypoxia, whereas the total amount of $NO_2^-$ and $NO_3^-$ did not differ from controls. In this study, $n = 7$ WT vessels were treated with $H_2O$; $n = 6$ WT vessels were treated with $Na_2S_2O_4$; and $n = 5$ WT vessels were treated with $N_2$ gas-treated buffer. For these experiments, $n = 5$ Hba1$^{+/+}$ mice, $n = 5$ Hba1$^{-/-}$ mice, $n = 6$ $Hba1^{fl/fl}$ mice, $n = 6$ EC $Hba1^{\Delta/\Delta}$ mice, $n = 4$ $Hba1^{WT/WT}$ mice, and $n = 4$ $Hba1^{WT/\Delta36-39}$ mice were used. Results for littermates were compared with one-sided $t$ tests. Bar graphs are centered on mean with error bars denoting standard error. Source data are provided in the Source data file.

genotype (Fig. 7h), thus excluding hydrogen peroxide as the vasodilatory signal in this context.

Critically, loss of vasodilation in the $Hba1^{-/-}$ and EC $Hba1^{\Delta/\Delta}$ groups is not dependent on the mechanism of hypoxia induction. When the buffer surrounding the vessels in the pressure myography setup was deoxygenated by bubbling with $N_2$ gas before use, dilation was induced in all genotypes except for the $Hba1^{-/-}$ and EC $Hba1^{\Delta/\Delta}$ groups in which endothelial alpha globin is absent (Fig. 7i). When $N_2$ gas was used, dilation could still not be inhibited by L-NAME (Fig. 7j) but was dependent on NO signaling because PTIO abolished dilation in all groups (Fig. 7k). Therefore, whether hypoxia was induced by pharmacologic treatment or by pre-treating the buffer with $N_2$ gas, dilation was still dependent on NO signals that correlated with the presence of endothelial alpha globin.

**Blood pressure change in response to environmental hypoxia is blunted in EC Hba1$^{\Delta/\Delta}$ mice**

Hypoxic vasodilation can also influence arterial blood pressure. Using radiotelemetry, we monitored the blood pressure of the $Hba1^{+/+}$, $Hba1^{-/-}$, $Hba1^{fl/fl}$, EC $Hba1^{\Delta/\Delta}$, $Hba1^{WT/WT}$, and $Hba1^{WT/\Delta36-39}$ mice. When exposed to room air (normoxia), the blood pressure of $Hba1^{-/-}$ and EC $Hba1^{\Delta/\Delta}$ mice was not significantly altered (Fig. 8a). However, in environmental hypoxia (10% $O_2$), the $Hba1^{+/+}$, $Hba1^{fl/fl}$, $Hba1^{WT/WT}$, and $Hba1^{WT/\Delta36-39}$ mice exhibited decreased systolic blood pressure as compared to their normoxic baseline, indicating a global vasodilatory response (as observed previously[25]) (Fig. 8b). The mean systolic blood pressure in $Hba1^{-/-}$ and EC $Hba1^{\Delta/\Delta}$ mice remained relatively unchanged (at 105 and 111 mmHg, respectively, in normoxia and 104 and 108 mmHg, respectively, in hypoxia), consistent with the observed

lack of hypoxic vasodilation response in isolated arteries. The differences between normoxic and hypoxic systolic blood pressures for the genotype groups are quantified in Fig. 8c. Differences in blood pressure between in individual animals are shown in Fig. 8d. Compared to littermate controls, mice with the $Hba1^{-/-}$ or EC $Hba1^{\Delta/\Delta}$ genotypes showed significantly less difference in blood pressure between normoxia and hypoxia. Concordantly, the heart rate of the $Hba1^{-/-}$ or EC $Hba1^{\Delta/\Delta}$ genotypes was not different when exposed to hypoxia, but other genotypes demonstrated a compensatory increase in heart rate to hypoxic exposure (Fig. 8e).

## Discussion

Hypoxic vasodilation as a result of NO generation from hemoproteins has been described previously[26]. Erythrocytic deoxygenated hemoglobin can act as a nitrite reductase to provide NO as a vasodilation signal to vessels. However, the circulatory location of this signaling (i.e., in resistance arteries vs. post-capillary venules[27]) and the efficiency with which erythrocyte-derived NO reaches vascular smooth muscle[5] remain undetermined. We hypothesized that endothelial alpha globin was uniquely situated to provide NO to vascular smooth muscle in hypoxic contexts. Our study identified a novel nitrite reductase function for endothelial alpha globin that may contribute to hypoxic vasodilation.

A role for alpha globin as a nitrite reductase presents an interesting juxtaposition with prior reports of alpha globin inhibiting NO signaling through interaction with eNOS and NO scavenging[9–11,22]. However, these two roles of alpha globin are distinct in the metabolic contexts in which they predominate. Alpha globin inhibition of eNOS-derived NO signaling necessarily requires $O_2$, as both eNOS enzyme

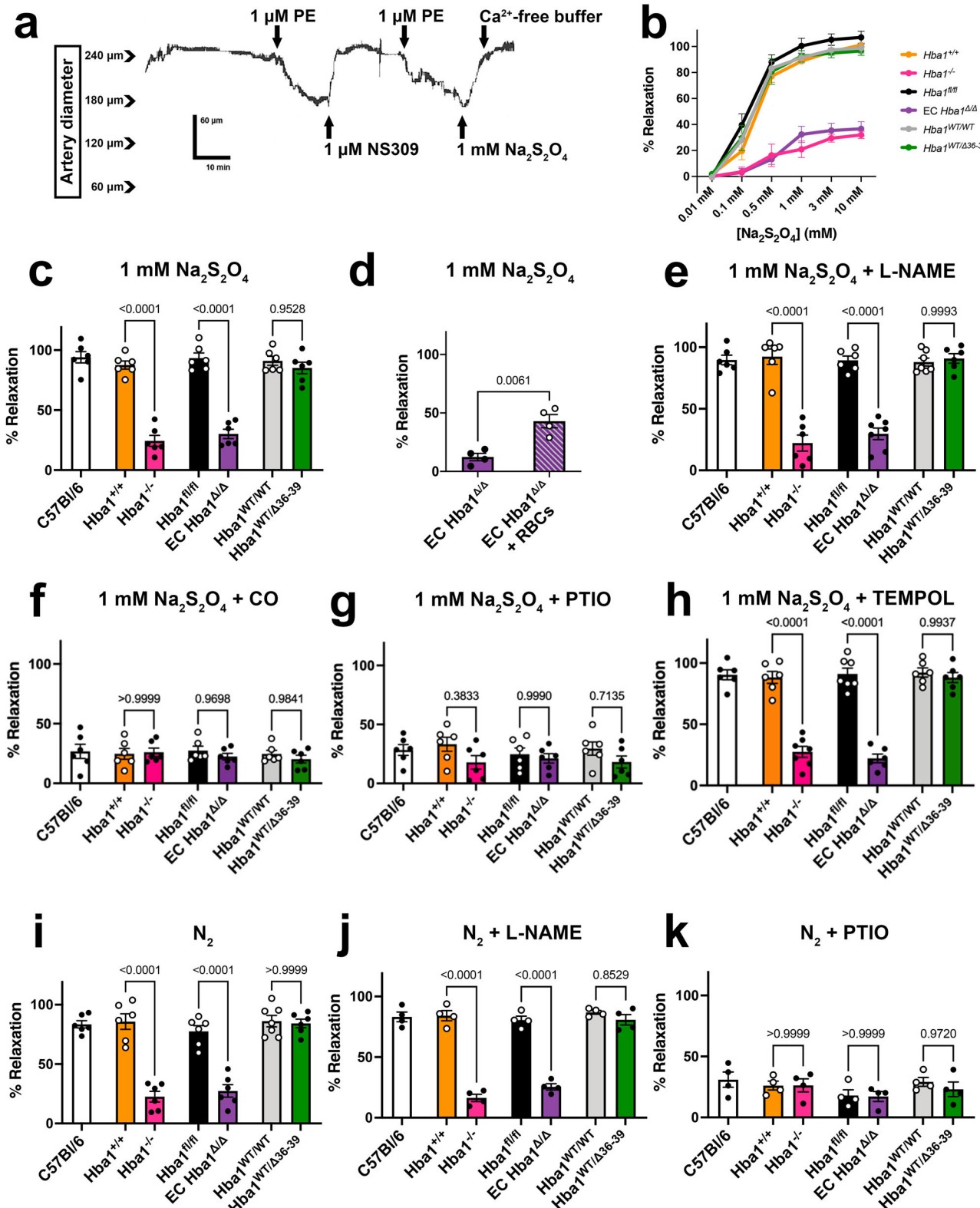

activity and NO scavenging and subsequent transformation to $NO_3^-$ by alpha globin require $O_2$. Conversely, endothelial alpha globin (as with erythrocyte hemoglobin) must be deoxygenated to act as a nitrite reductase[27]. Therefore, the opposed roles for alpha globin in NO signaling are separated by local metabolism: when $O_2$ is prevalent, alpha globin may predominate as an NO signaling inhibitor; however, when tissues are metabolically active and $O_2$ is scarce, endothelial alpha globin can act as a hypoxia sensor and signal enactor to increase

perfusion and match $O_2$ supply to metabolic demand. We have demonstrated that isolated alpha globin can reduce nitrite to NO in hypoxic conditions and that this reaction generates NO in thoracodorsal artery lysate (Fig. 2). The nitrite reductase activity of isolated alpha globin is almost 3 times faster than that of tetrameric HbA (Fig. 2b). Importantly, this reaction probably occurs where alpha globin is localized at the myoendothelial junction, a specialized signaling domain where endothelial and smooth muscle cytoplasms are in close

**Fig. 7 | Hypoxic vasodilation is decreased without endothelial alpha globin.**
**a** Representative trace of a hypoxic vasodilation experiment. Cannulated vessels were treated with an initial dose of phenylephrine (PE) to induce constriction, and endothelial dilation integrity was confirmed by dilation in response to NS309. A further PE dose was then administered to induce constriction, and dilation in response to $Na_2S_2O_4$ was assessed. Finally, maximum dilation to $Ca^{2+}$-free buffer established the percentage of the maximum dilation observed for each treatment. **b** Vasodilation in response to increasing doses of $Na_2S_2O_4$ was blunted by the global loss of *Hba1* (Hba1$^{-/-}$, orange) and endothelial deletion of alpha globin (EC *Hba1*$^{Δ/Δ}$, purple), but not by decreasing the interaction of alpha globin and eNOS (*Hba1*$^{WT/Δ36-39}$, green). **c** Hypoxia induced by a single dose of 1 mM $Na_2S_2O_4$ blunted dilation only in mice with the global Hba1$^{-/-}$ and EC *Hba1*$^{Δ/Δ}$ genotypes. The hypoxic vasodilation was preserved in *Hba1*$^{+/+}$, *Hba1*$^{fl/fl}$, *Hba1*$^{WT/WT}$, and *Hba1*$^{WT/Δ36-39}$ groups. **d** Adding blood to the lumen of the EC *Hba1*$^{Δ/Δ}$ vessels partially restored dilation after $Na_2S_2O_4$ treatment. **e** The hypoxic vasodilation response was not affected by the pretreatment of the vessel with L-NAME, a NOS inhibitor. **f** The hypoxic vasodilation response of the

isolated resistance arteries was inhibited by carbon monoxide. **g** The dilation in response to $Na_2S_2O_4$ is NO-mediated, as treatment with the NO scavenger PTIO (2-phenyl-4, 4, 5, 5,-tetramethylimidazoline-1-oxyl 3-oxide) prevented dilation in all groups. **h** Limiting other reactive oxygen species (including superoxide and hydrogen peroxide) with TEMPOL did not restore a dilatory response to chemical hypoxia in the *Hba1*$^{-/-}$ or EC *Hba1*$^{Δ/Δ}$ groups. **i** Treating the buffer surrounding the vessels with $N_2$ gas to deoxygenate enabled hypoxic vasodilation in all groups in which endothelial alpha globin is present, and this effect is not diminished with NOS inhibition through L-NAME pretreatment **j**. **k** The response to $N_2$ gas is also NO dependent, as it was inhibited by PTIO treatment. All experiments in **b**, **c**, and **f–i** were performed with 6 animals per genotype. The experiment in **d** was with 4 animals. Experiments in **j** and **k** were with 4 animals per genotype. All comparisons between groups used two-sided *t*-tests with multiple comparisons correction, except for the study in **d**, for which a *t* test was used. Bar graphs are centered on mean with error bars denoting standard error. Source data are provided in the Source data file.

apposition. This highly specialized reaction localization could increase the relative efficiency of NO generation through alpha globin nitrite reduction, leading to an increased vasodilatory effect. However, once each alpha globin monomer has completed the nitrite reduction reaction, the heme iron must be recycled from a met-heme or ferric state to a ferrous state to bind another gas molecule or reaction with $NO_2^-$ [40]. This process has been demonstrated through heme reduction by cytochrome b5 reductase 3[11]. The reductase domain of eNOS[28], or other reducing equivalents, including ascorbate[29], could also contribute to heme recycling.

To determine whether alpha globin could control vasodilation through nitrite reduction separately from its role in eNOS-derived NO scavenging, we used two novel loss-of-function mouse models to demonstrate that endothelial alpha globin participates in hypoxic vasodilation through an autonomous nitrite reductase mechanism. First, an endothelial-specific deletion of alpha globin (EC *Hba1*$^{Δ/Δ}$) demonstrated the effects of loss of the eNOS-inhibition function and nitrite reductase function (through the heme moiety) of endothelial alpha globin. Second, the global mutation of alpha globin through the deletion of four residues (*Hba1*$^{Δ36-39}$) prevented the association of alpha globin and eNOS, thereby decreasing NO scavenging[9,10], while preserving nitrite reductase activity. By combining these two models, we could specifically assay the nitrite reductase role of endothelial alpha globin.

First, the distance to exhaustion as assayed by the treadmill running test was decreased in mice with the EC *Hba1*$^{Δ/Δ}$ genotype, as compared to littermate *Hba1*$^{fl/fl}$ controls (Fig. 5). This difference contrasts with that in the *Hba1*$^{WT/WT}$ and *Hba1*$^{WT/Δ36-39}$ comparison, in which the mutant animals and littermate controls were exhausted at a similar distance. Additionally, in the global *Hba1*$^{-/-}$ genotype, the distance to exhaustion was also decreased, although this was confounded by the anemia in those mice (Supplementary Fig. 6 and Supplementary Fig. 12). The running distance appeared to differ between *Hba1*$^{fl/fl}$ genotypes and *Hba1*$^{WT/Δ36-39}$ genotypes, although this discrepancy could reflect strain differences[30].

Second, vascular cell nitrite consumption appeared to be significantly decreased in EC *Hba1*$^{Δ/Δ}$ mice (Fig. 6). Incubating isolated thoracodorsal arteries with $Na_2S_2O_4$ resulted in WT vessels having significantly decreased nitrite. Vessels isolated from EC *Hba1*$^{Δ/Δ}$ mice had nitrite levels similar to those in untreated vessels, even after 5 min of $Na_2S_2O_4$ treatment. There was a significant increase in cellular nitrite in EC *Hba1*$^{Δ/Δ}$ mice as compared to littermates, whereas the levels were similar in the *Hba1*$^{Δ36-39}$ genotypes. Additionally, EC *Hba1*$^{Δ/Δ}$ mice had decreased $NO_3^-$ levels with unchanged total ($NO_2^- + NO_3^-$) amount. Overall, cellular nitrite consumption in the thoracodorsal artery is significantly affected by the presence of endothelial alpha globin.

Finally, a dilatory response to hypoxia is absent in EC *Hba1*$^{Δ/Δ}$ vessels, where endothelial alpha globin is absent. Loss of the entire

alpha globin protein in EC *Hba1*$^{Δ/Δ}$ or global *Hba1*$^{-/-}$ resistance vessels largely prevented dilation in response to hypoxic buffer conditions (Fig. 7b). However, targeted deletion of the eNOS-interacting motif did not affect the vasodilation response to hypoxia. We determined that the vasodilatory response was independent of both NOS and hydrogen peroxide but requires NO. Furthermore, the dilation in response to $Na_2S_2O_4$-induced hypoxia is dependent on a CO-inhibitable process, consistent with heme-based nitrite reduction. Integrating the dilatory response of individual arteries with whole animal physiology, the normal decrease in blood pressure associated with acute exposure to arterial hypoxia was abrogated in mice lacking endothelial alpha globin expression (Fig. 8c, d). The compensatory increase in heart rate after exposure to hypoxia, as observed in control genotypes, was not observed in the *Hba1*$^{-/-}$ or EC *Hba1*$^{Δ/Δ}$ genotypes. Because the blood pressure of the mice in *Hba1*$^{-/-}$ or EC *Hba1*$^{Δ/Δ}$ groups did not change drastically, the heart rate was not significantly increased in those groups (Fig. 8e) Differences were observed in these experiments only when endothelial alpha globin was deleted and was, therefore, unable to participate in nitrite reduction.

We acknowledge some limitations in these studies. First, the *Hba1*$^{Δ36-39}$ model is complicated by the homozygous lethality; therefore, heterozygous adults were necessarily used in the experiments on adult animals. Additionally, this mutation is constitutive and global, precluding observations of endothelial alpha globin specifically. However, using littermate controls in all studies enabled phenotypic comparisons of WT and heterozygous disruption of alpha globin/eNOS interaction. Second, many of our experiments were performed in a whole-animal physiology context, in which erythrocytes can contribute to hypoxic vasodilation. Although our EC *Hba1*$^{Δ/Δ}$ mutation was targeted to endothelium using the *Cdh5-Cre*$^{ERT2}$, we did not specifically measure the contribution of erythrocytes to NO signaling in exercise hyperemia or hypoxic blood pressure compensation. The use of isolated skeletal muscle resistance arteries from each of our genetic contexts demonstrated that endothelial alpha globin can induce vasodilation in skeletal muscles, but the proportions of endothelial alpha globin or erythrocytic hemoglobin signaling were not directly measured.

Overall, our experiments demonstrate that endothelial alpha globin can participate in hypoxic vasodilation through nitrite reduction. Alpha globin has an optimal expression localization that is critical for its function in controlling hypoxic vasodilation. First, endothelial alpha globin expression is normally restricted to resistance vasculature, specifically arterioles[11]. Therefore, as a nitrite reductase, alpha globin can provide NO signaling where it is maximally effective to induce dilation and changes in tissue perfusion. Dilation of the precapillary arterial beds increases tissue perfusion without the need for signals to propagate up or down the vascular tree. Second, the subcellular localization of alpha globin to the MEJ[11,22] could provide

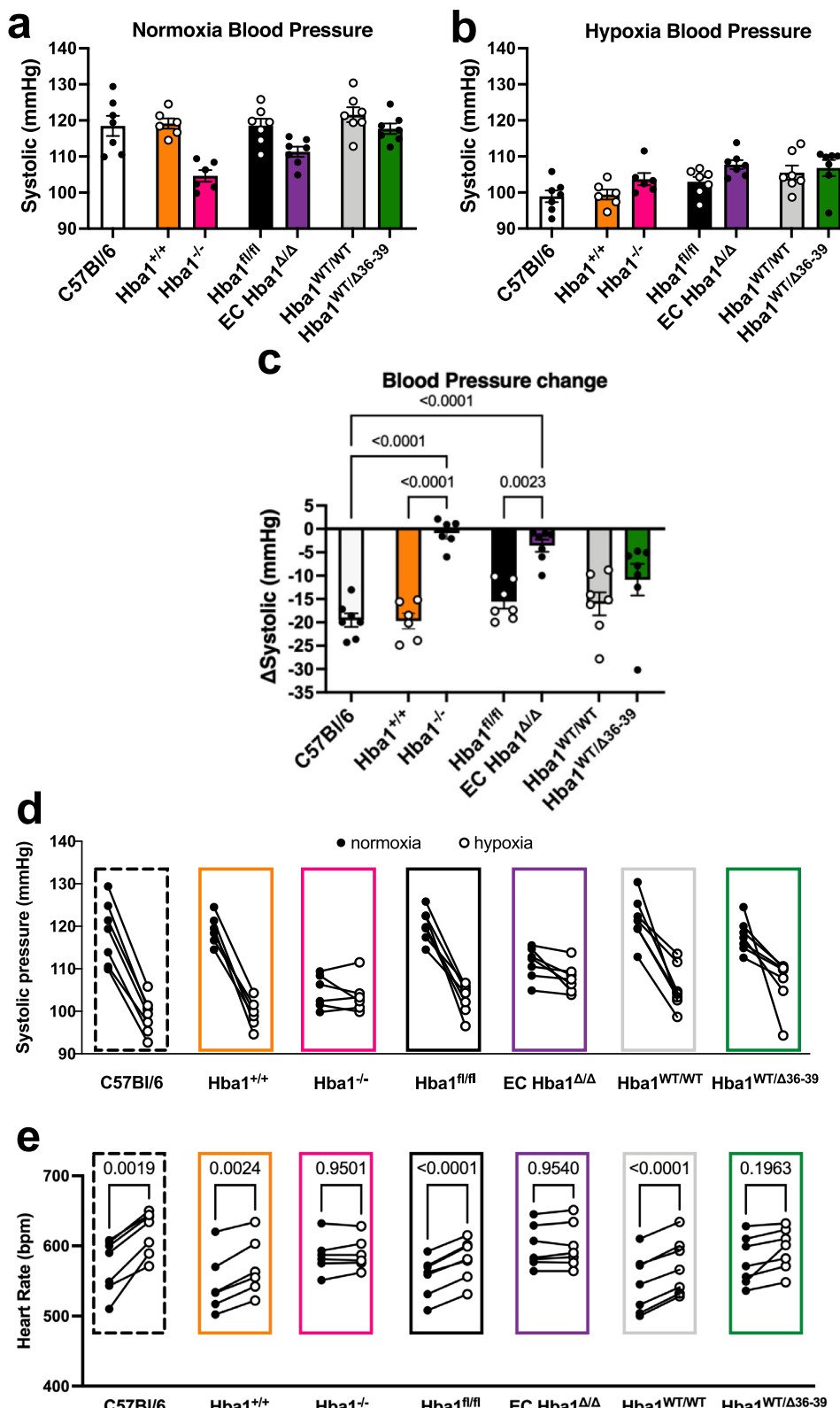

additional benefits in terms of the spatial optimization of NO generation. The MEJ is a domain of the endothelium that is in direct contact with underlying vascular smooth muscle[31]; therefore, NO generation by MEJ-restricted alpha globin minimizes the diffusion distance necessary to enact vasodilation through smooth muscle relaxation.

This novel role for endothelial alpha globin places alpha globin at a fulcrum of NO signaling in the resistance vasculature. As both an inhibitor of eNOS-produced NO and a source of NO generation in hypoxia, alpha globin uniquely determines resistance vascular function in the endothelium.

## Methods

All research complies with NIH guidelines and the rules set forth by the University of Virginia Institutional Animal Care and Use Committee

**Fig. 8 | Endothelial alpha globin enhances blood pressure change in response to global hypoxia. a** In normoxia (room air), the systolic blood pressure of the *Hba1*[−/−] and EC *Hba1*[Δ/Δ] genotypes was slightly decreased when compared to other genotype groups. **b** When exposed to hypoxia, the *Hba1*[−/−] and EC *Hba1*[Δ/Δ] groups had slightly elevated systolic blood pressure when compared to other groups. **c** *Hba1*[−/−] and EC *Hba1*[Δ/Δ] mice showed the least change in systolic blood pressure difference between normoxia and hypoxia when compared to littermate controls and WT mice. This difference between groups was statistically significant. **d** Data points from individual animals plotted to show the change in systolic pressure from

normoxia (filled circles) to hypoxia (open circles). **e** Heart rate in beats per minute (bpm) of the individual animals used for this experiment are plotted to show change when the animal was exposed to hypoxia. For these experiments, $n = 7$ C57Bl/6 mice, $n = 6$ *Hba1*[+/+] mice, $n = 6$ *Hba1*[−/−] mice, $n = 7$ *Hba1*[fl/fl] mice, $n = 7$ EC *Hba1*[Δ/Δ] mice, $n = 7$ *Hba1*[WT/WT] mice, and $n = 7$ *Hba1*[WT/Δ36−39] mice were used. Comparisons across groups used a one-way ANOVA; in **e**, only comparisons between paired values were calculated. Bar graphs are centered on mean with error bars denoting standard error. Source data are provided in the Source data file.

and Institutional Review Board. Human studies were carried out in adherence to the principles of the Declaration of Helsinki.

## Animals

*Hba1*[fl/fl] mice were developed in partnership with Ingenious Targeting Laboratory. *Hba1*[fl/fl] mice were crossed with the *Cdh5-PAC-Cre*[ERT2] transgenic driver to yield a model of tamoxifen-inducible, endothelial cell-specific *Hba1* deletion. EC Hba1[Δ/Δ] mice and Cre (−) littermate controls received 10 days of tamoxifen injections (100 µL at 10 µg/mL) starting at 6 weeks of age. Both male and female mice were used for experiments. This line was backcrossed onto a C57Bl/6 for at least 8 generations. Mice were maintained on Teklad irradiated rodent diet (Teklad product no. 7912). All animals were used between 12 and 20 weeks of age. All animal experiments described in this study were approved by the University of Virginia IACUC and were carried out in adherence to the relevant ethical guidelines.

## Human samples

The collection and use of human adipose tissue biopsies were approved by the University of Virginia's Institutional Review Board for Health Sciences Research (Study 17194). Human volunteers had an approximate 40 mm×20 mm biopsy of adipose tissue removed from the abdomen, which was immediately placed in ice-cold KREBS buffer and arterioles dissected manually within 30 min of removal, placed in 4% paraformaldehyde, and paraffin embedded. Informed consent that de-identified samples would be used for research purposes was obtained from all participants enrolled in the IRB-approved study.

## Generation of CRISPR/Cas9 constructs

*HBA-a1* (NC_000077.6) targeting site for guide RNAs were identified using the online ZiFiT (Zinc Finger Targeter) software version 4.2 (http://zifit.partners.org)[32]. The sgRNA target site 5′- GGAAAGT AGGTCTTGGTGGT −3′ and its recommended 22 nucleotide long oligos(oligo 1: 5′-TAGGAAAGTAGGTCTTGGTGGT-3′; oligo 2: 5′-AAAC ACCACCAAGACCTACTTT-3′) were chosen from ZiFiT output. Oligo1 and oligo 2 were annealed and cloned at a *Bsa*I site in the DR274 vector (http://www.addgene.org/crispr/jounglab/crisprzebrafish/) then used to transform in chemically competent XL-1 Blue cells. Antibiotic-resistant positive clones were checked by sequencing with M13 primers. Positive PDR274 clones were digested with *Dra*I restriction enzyme (New England Biosciences) and used as templates for in vitro transcription using a MEGAshortscript T7 Transcription Kit (Life Technologies). sgRNAs were DNAse treated and purified using a MEGAclear Kit (Ambion cat. no. AM1908).

Production of Cas9 nuclease mRNA used the pMLM313 expression vector (Addgene plasmid #42251), which harbors the T7 promoter site upstream of the translational start site, and a nuclear localization signal at the C-terminus. The pMLM313 plasmid was linearized using a *Pme*I restriction enzyme (New England Biosciences), separated by agarose gel eletrophoresis, and extracted from the gel with a NucleoSpin Extract II kit (Clontech cat. no. 636972). The linearized and purified pMLM313 fragment was used as a template for in vitro transcription of Cas9 mRNA with a mMESSAGE mMACHINE T7 ULTRA kit (Life Technologies)[33,34]. After transcription, Cas9 mRNA underwent poly-A tailing reactions and DNAse I treatment according to the

manufacturer's instructions. The concentration was determined using the NanoDrop instrument (Thermo Scientific).

## Microinjection

Pronuclei of single-cell stage embryos were collected from super-ovulated donor female mice (B6.SJL strain) and microinjected with CRISPR/Cas9 constructs by using micromanipulators and injection needles (World Precision Instruments, filament #1B100-6). Injected embryos were implanted into pseudo-pregnant recipient female mice in accordance with NIH guidelines. Two-week-old pups born to recipient mice were screened by PCR, using *Hba* gene-specific primers, and sequenced for the presence of CRISPR/Cas9-induced mutations. The *Hba* gen-FP primer sequences were as follows: forward: 5′- ATATGGACCTGGCACTCGCT −3′ and reverse: 5′- GTCCCAGCGCAT ACCTTGAA −3′. These mice were maintained as a mixed-strain background through inbreeding.

## Whole-genome sequencing for SNP and Indel analysis

Whole-genome sequencing was performed by GeneWiz (South Plainfield, NJ) using contiguous amplicon sequencing on an Illumina MiSeq instrument in a $2 \times 150$-bp paired-end configuration. DNA library preparation without fragmentation, multiplexing, sequencing, unique sequence identification, and relative abundance calculations were performed in house at GeneWiz. Sequence reads for each sample were aligned to the *Mus musculus* genome (GRCm38, as reference sequence) after trimming the adapters and nucleotides with poor quality. SNPs/INDELs were detected using the probabilistic model. The settings of parameters for analysis: minimum frequency = 25%; minimum coverage = 10; minimum count of a variant = 4. The detected variants were annotated for amino acid changes and against known mutations of *M. musculus*. The mapping rate for each sample was >95%. Three animals were used as data points for these analyses; animals genotyped to be heterozygous were compared against the animal WT for the mutation. A summary of the annotated SNP/INDEL analysis can be found in the Source data File. Targeted sequencing reads from the mutation site have been deposited to NCBI BioProject # PRJNA870663.

## Blood cell measurements

Isolated mouse blood was analyzed by the St. Jude Children's Hospital Blood Pathology Labs as described previously[9] to determine hematocrit, hemoglobin content, and the number of red cells. Additionally, reticulocytes and red cell size were determined using automated methods.

## Immunofluorescence

Thoracodorsal arteries were excised, cleared of blood, and fixed overnight in 4% paraformaldehyde. After being washed with ethanol, vessels were embedded in agarose and then in paraffin blocks and sectioned at 5 µm thickness. Paraffin was removed by a Histoclear bath, and sections were rehydrated through a 100%/95%/70% ethanol/water gradient before antigen retrieval in citrate solution (pH 6). PBS blocking solution (including 0.25% Triton X-100, 0.5% bovine serum albumin, and 5% donkey serum) was used to block sections and as a primary and secondary antibody diluent. Primary antibodies were rabbit anti-hemoglobin alpha (abcam, #102758, 1:250 dilution), rabbit

anti-hemoglobin beta (abcam, #227552, 1:250 dilution), rabbit anti-myoglobin (abcam, #77232, 1:250 dilution), rabbit anti-neuroglobin (abcam, #197670, 1:250 dilution), rabbit anti-cytoglobin (abcam, #202972, 1:250 dilution), and mouse anti-eNOS (BD Bioscience, #610296, 1:500 dilution). Secondary antibodies used were donkey anti-rabbit (Alexa Fluor 647, Life Technologies A-31571) and donkey anti-mouse (Alexa Fluor 594, Life Technologies A-21207). Stained sections were stained with DAPI, mounted, and imaged using an Olympus FV1000 confocal microscope with Olympus Fluoview FV1000 software using DAPI, and 488 nm (to visualize autofluorescence of the elastic lamina), 594 nm, and 647 nm channels for fluorescence detection.

### Detection of globin precipitates from red blood cells

Globin precipitates from erythrocytes were analyzed as described previously[35–39]. Briefly, washed red blood cells (RBCs) were lysed, the lysate was centrifuged, and the pellets were washed extensively in ice-cold 0.05× PBS. Membrane lipids were extracted with 56 mM sodium borate, pH 8.0, with 0.1% Tween-20 at 4 °C. Precipitated globins were dissolved in 8 M urea, 10% acetic acid, 10% b-mercaptoethanol, and 0.04% pyronin, fractionated by triton-acetic acid-urea (TAU) gel electrophoresis, and stained with Coomassie blue.

### Fluorescence polarization

Fluorescently labeled peptides were synthesized by AnaSpec. The fluorescence polarization was performed as described previously[9] using a SpectraMaxM5 plate reader and SoftMax software (version 5). Briefly, 1 μM recombinant oxygenase domain of eNOS was incubated with a peptide composed of the alpha globin binding sequence (LSFPTTKTYF) or the same peptide lacking Hba1 amino acids 36-39 (sequence: LTKTYF) to assess the binding affinity. For this assay, each peptide was synthesized with an Alexa Fluor 488 tag (excitation: 494 nm, fluorescent lifetime: 4.1 ns) at the C-terminus.

### Proximity ligation assay

Proximity ligation assays (PLA) were performed as described previously[11]. Sections (as in "Immunofluorescence" subsection) were stained with the same primary antibodies at the same concentrations. DUOLINK® PLA probes, ligase, and polymerase were all used in accordance with the manufacturer's instructions. Sections were mounted, stained with DAPI, and imaged with DAPI, Alexa 488, and Texas Red (for the PLA fluorophore) channels. Images of two sections per animal (blinded as to genotype) were quantified by tracing the outline of the internal elastic lamina (IEL) in ImageJ and dividing the number of puncta by the total perimeter of the elastic lamina.

### Immunoblots

Immunoblots were quantified against total protein by using a LI-COR Odyssey imaging system with near-infrared fluorescent secondary antibodies as previously described[40]. Specific antibodies used are listed in supplemental materials.

### Nitric oxide imaging

Arterioles were surgically opened and pinned on a Sylgard block in *en face* preparation. NO levels were assessed using 5 μM DAF-FM DA (4-amino-5 methylamino-2',7'-difluorofluorescenin diacetate) prepared in HEPES-PSS (physiologic saline solution) with 0.02% pluronic acid[41]. DAF-FM DA reacts with NO in the presence of oxidants and forms a fluorescent triazole compound. En face mesenteric arteries were pre-treated with 10 μM carbachol (CCh, a muscarinic receptor agonist) in HEPES-PSS for 5 min at 30 °C. The arteries were then incubated with DAF-FM containing CCh for 20 min at 30 °C in the dark. DAF-FM-DA fluorescence was imaged using an Andor Revolution WD (with Borealis) spinning-disk confocal imaging system (Andor Technology, Belfast, UK), consisting of an upright Nikon microscope with a water immersion objective (60× magnification, numerical aperture [NA] 1.0)

and an electron-multiplying CCD camera. Images were obtained along the z-axis at a slice size of 0.05 μm from the top of the endothelial cells to the bottom of the smooth muscle cells, using an excitation wavelength of 488 nm, and emitted fluorescence was captured with a 525/36-nm band-pass filter. DAF-FM fluorescence was analyzed using custom-written SparkAn software designed by Dr. Adrian D. Bonev (University of Vermont, Burlington, VT). An outline was drawn around each endothelial or smooth muscle cell to obtain the arbitrary fluorescence intensity of that cell. The background fluorescence was then subtracted from the recorded fluorescence. The fluorescence from all the cells in a field of view was averaged to obtain a single fluorescence number for that field. Relative changes in DAF-FM fluorescence were calculated by dividing the fluorescence in the treatment group by that in the control group. Each artery was considered as $n = 1$. Several fields of view from each artery were averaged to get one averaged fluorescence value for that artery. Thus, $n = 3$ represents data from 3 arteries.

### Treadmill running test

Mice aged 10–13 weeks were acclimatized to the treadmill in accordance with established methods[42]. On the day of testing, the treadmill was set to a 5% incline at a speed of 13 m/min (0.5 miles per hour) for 30 min, and the speed was increased by 2.7 m/min (0.1 miles per hour) every 30 min (to a maximum of 27 m/min [1.0 miles per hour]) until the mice reached exhaustion. Exhaustion was defined as a refusal to run despite receiving 20 strokes on the tail with a wire brush. Blood lactate was measured immediately before and after the test via tail snip and lactate measurement using a Lactate Scout handheld monitor. All mice were monitored by an investigator blinded to their genotypes.

### Nitrite/nitrate measurement

Vessels were blotted dry and immersed in a 1.5 mL microcentrifuge tube containing 50 μL PBS with 2.5 mM EDTA and 100 mM NEM, pH 7.4. Vessels were sonicated twice with a Sonic Dismembrator Model 100 (Fisherbrand) for 10 s per cycle (0.05 watts). Then, 100 μL of methanol was added and samples were vortex mixed for 20 s. After centrifugation at $19,000 \times g$ for 10 min at 4 °C, 100 μL of each supernatant was collected and the nitrite and nitrate were measured using an NOX analyzer (ENO-30). Concentrations were calculated by comparison to freshly prepared sodium nitrite and sodium nitrate standard curves.

### Measurement of nitrite reductase activity of alpha globin

Human alpha globin (4 μM) or hemoglobin A (4 μM) were reacted with sodium nitrite (0.25–4 mM) in the presence of 1 mM sodium dithionite in 0.1 M sodium phosphate buffer, pH 7.4, at 25 °C. Optical spectra were recorded using a Cary 60 spectrophotometer fitted with temperature control. Rate constants were determined from the time course of spectrum change (430 nm–415 nm) by fitting to exponential functions using SigmaPlot 10.0.

### Measurement of NO production by isolated vessels

This experiment was performed according to the setup described previously in Totzeck et al[43]. Briefly, NO production under anaerobic conditions was measured by chemiluminescence.

### Vasoreactivity

Cumulative dose-response curves for pressurized and cannulated thoracodorsal arteries (140–200 μm in diameter), as previously described, were used throughout the studies. Briefly, freshly isolated thoracodorsal arteries were placed in ice-cold Krebs-HEPES solution containing 118.4 mM NaCl, 4.7 mM KCl, 1.2 mM MgSO$_4$, 4 mM NaHCO$_3$, 1.2 mM KH$_2$PO$_4$, 2 mM CaCl$_2$, 10 mM HEPES, and 6 mM glucose. The vessels were then mounted in a pressure arteriograph (Danish Myo-Technology) with their lumens filled with Krebs-HEPES solution. The vessels were pressurized to 80 mmHg and heated to 37 °C. Between

experiments, the bath solution was washed out for 10 min with fresh Krebs-HEPES buffer, after which a new Krebs-HEPES solution was added and allowed to re-equilibrate. All vessels were tested for endothelial function before the $NaS_2O_4$ experiments by assessing dilatory response to 1 µM NS309. The phenylephrine preconstriction diameter was determined by adding 1 µM phenylephrine to the bath, and waiting the internal diameter of the vessel reached a stable plateau before proceeding to the next stimulation condition. After the conclusion of the experimental stimulations, the maximum constriction of the vessels was determined by stimulation with 3 mM KCl. Finally vessels were washed with a $Ca^{2+}$-free Krebs- HEPES solution supplemented with 1 mM EGTA and 10 µM sodium nitroprusside to determine their maximum passive diameter. Dilation in response to $NaS_2O_4$ or bubbled $N_2$ gas was calculated as (diameter after $NaS_2O_4/N_2$ stimulation − diameter before $NaS_2O_4/N_2$ stimulation) ÷ (passive diameter − diameter before $NaS_2O_4/N_2$ stimulation) × 100. The software used to analyze this data was also from Danish MyoTechnology, MyoView v5.

### Blood pressure

Radiotelemetry was used to monitor blood pressure and heart rate as previously described, using Ponemah v6[9]. Acute exposure to 10% $O_2$ (30 min) occurred in modified hypoxia chambers on telemetry platforms. There were no differences in diurnal blood pressure between genotypes.

### Statistics

In all cases, mean values are shown with error bars denoting the SEM unless otherwise stated. Statistical tests are denoted in the figure captions below each figure. GraphPad Prism v9 was used to analyze statistical relationships between groups.

## Data availability

The data that support the findings of this study are available in the manuscript, Source data file, and accompanying Supplementary Information. Additional data are available from the corresponding author. Source data are provided with this paper.

## Code availability

The SparkAn software was a kind gift of Dr. A Bonev (University of Vermont) and can be accessed directly: https://github.com/vesselman/SparkAn

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

## Acknowledgements

The authors thank Dr. Laurent Kiger for useful discussions, and Keith A. Laycock, PhD, ELS (St. Jude Department of Scientific Editing) for scientific editing of the manuscript. For help in performing the experiments on nitrite reduction, we would like to acknowledge Junjie Li, PhD, Anthea LoBue, MSci, and Silvia Gardeweg, DVM. Some artwork was illustrated for this article by Anita Impagliazzo. This work was supported by NIH HL088554 (BEI and MW), GM100776 (TPB), HL098032 (DBKS), R35 GM131829 (LC), F31HL1440032 (TCSK). This research was supported in part by the Intramural Research Program of the NIH, NIAID project number AI001150 (SB and HCA). The content is solely the responsibility of the authors and does not necessarily represent the official views of the National Institutes of Health.

## Author contributions

T.C.S.K. performed a component of most of the experiments, including (not limited to) immunocytochemistry, PLA, genetic analysis and FP, vasoreactivity and NO biochemistry, as well as write drafts of the manuscript. A.I. and C.A.R. performed immunocytochemistry and vasoreactivity. L.J.D. performed immunocytochemistry and genetic analysis. A.K.B. and E.M. performed mouse husbandry and expanded/tested the Hba1 floxed mouse and Hba1$^{\Delta 36-39/WT}$ mouse lines. E.M., Z.Y.T., H.R.A.P., J.T.B., and G.B.B.-F. performed vasoreactivity and blood pressure experiments. C.L., M.M.C.-K., K.R., and R.P. performed NO biochemistry experiments. A.S.K. performed mouse exercise and vasoreactivity experiments. P.S. and W.X. created the Hba1$^{\Delta 36-39/WT}$ mouse line. K.H. and S.K.S. performed en face DAF measurements. L.C., M.J.W., T.P.B., Z.Y., G.C., D.B.K.-S., A.S.P., A.S.K., S.B., M.M.C.-K., R.P., and H.A. provided experimental design, data interpretation, and writing. B.E.I. and T.C.S.K. conceived the ideas behind the work, data interpretation, and most of the experimental design. B.E.I. wrote and financed the work.

## Competing interests

The authors declare no competing interests.

## Additional information

**Correspondence and requests** for materials should be addressed to Brant E. Isakson.

