## [Peer Review File · Nature Communications]

Endothelial alpha globin is a nitrite reductaseREVIEWER COMMENTS

Reviewer #1 (Remarks to the Author):

This is an interesting and potentially important work. I have some points that for the authors to consider.

Is there an overreliance on dithionite in these studies? It is a strong reductant that may change the oxidation state of many biomolecules (and not simply reduce molecular oxygen), including those that may be important for control of vasotone and blood pressure. Even the product formed during dithionite oxidation may, induce oxidation of targets to have an impact on vasodilation. Therefore, can some experiments with authentic hypoxia be performed? Does dithionite in a test tube directly react with nitrite, or in cells is it possible there is some catalysis of this direct reaction?

Figure 4C – why is the HBa fl/fl so different than the Hba WT/WT.

Figure 4D – how can lactate decrease upon exercise in some animals. It seems the EC Hba null shows this loss lactate phenotype more prevalently? Is this a real biological event and if so, what might it mean?

Figure 4e – capillary density is double in the Hba WT/delta36-39 mice. What does this mean?

Figure 5E – “NOx” is mentioned – does that include low molecular weight and protein nitrosothiols?

Why is the dithionite experimental groups in Figure 5 not also shown under aerobic conditions?

Figure 6B – do the profiles of vasodilation look the same if normalization is not used and the force data are plotted? It seems from some other data presented in previous Figures that the two transgenics lines have alterations to their vasculature and I am wondering if there are differences in the dilatory or constrictor responses of these vessels that is not apparent due to normalization.

As mentioned above, it would be good if the data with authentic hypoxia (not chemical hypoxia) can be provided – for example in Figure 6B.

Figure 6h utilizes TEMOL – often described as an SOD mimetic. If this is the case, one would expect the production of hydrogen peroxide (a vasoactive molecule) at a higher rate when it is added. What do the authors think about this?

I think it is very important that the in vivo blood pressure measurements are replicated a lot more, having observed treatment effects with low samples sizes. The diurnal data would be of interest, as would the diastolic and mean arterial values. How does heart rate and cardiac output change between genotypes and with hypoxia? For example, are there changes to these parameters that complicate the interpretation of the systolic blood pressure changes reported?

Reviewer #2 (Remarks to the Author):

GENERAL COMMENTS

Keller and colleagues report studies of the dual functions of endothelial cell (EC) Hba1 related to nitrite reduction and eNOS-derived NO scavenging. They employ mice with EC-specific deletion of Hba1 and mice with mutant Hba1 incapable of inhibitory association with eNOS. The studies were carefully carried out using standard methods, and they demonstrate that EC Hba1 is required for hypoxia-related vasodilation in isolated arteries, and for the hypoxia-related decline in systolic BP. The reported observations regarding EC Hba1 and nitrite consumption by hypoxic isolated vessels

are also convincing, but the data would be strengthened if it included results for vessels in all genotype groups studied under normoxic conditions. The changes in exercise tolerance observed in mice with EC Hba1 deletion are modest. The primary limitation of the work is that firm conclusions cannot be drawn about EC Hba1 NO scavenging because fetal lethality required that the Hba1 mutant mouse studies be done in heterozygous mice. Other concerns are raised about data presentation and interpretation.

SPECIFIC COMMENTS

Abstract

1. It should be more clearly mentioned in the abstract that the studies of the mutant mice were done in heterozygotes.

Methods

2. Why were the studies of hypoxia-related vasodilation limited to the use of chemical hypoxia? This should be stated.

Results

3. What was the sample source for the PCR amplification studies shown in Fig. 2B?

4. Fig. 2C- the location of EC Hba1 at myoendothelial junctions is not convincing (and there are no related breaks in the internal elastic lamina). This feature needs to be better visualized in order to make this claim.

5. Suppl. Fig. 5C- what tissue was imaged? This should be mentioned at least at a minimum in the legend.

6. Suppl. Fig. 6B- treatment groups should be labeled in the graph.

7. Fig. 4C- why is there a difference in distance to exhaustion in the two control groups (Hba1 f/f versus Hba1 wt/wt)?

8. Line 388-390- It is not possible to conclude that preventing Hba1-eNOS association does not affect exercise capacity because the loss-of-function is not complete.

9. Fig. 5- data are needed for H₂O-only treated arteries in each genotype group so changes in nitrite consumption with chemical hypoxia can be evaluated.

10. Fig. 6- the authors need to provide evidence of effective antagonism of eNOS-dependent vasodilation with L-NAME in the specific model system employed.

11. Fig. 6H, and line 433- instead of "baseline dilation...", should the sentence read "hypoxic vasodilation...?"

Discussion

12. Lines 478-479- Although limitations of the work are appropriately mentioned later in the Discussion, here "we are able to specifically assay a nitrite reductase role for endothelial Hba1" is an overstatement.

13. Line 486- differences in running distance between Hba1 fl/fl controls and Hba1 wt/wt controls should be mentioned here.

Reviewer #3 (Remarks to the Author):

Keller and colleagues have undertaken an extensive and well thought out series of experiments to explore the role of "Endothelial alpha globin as a nitrite reductase". This work is based upon the initial observation by Straub and Isakson in 2012 that alpha globin is expressed in the myoendothelial junction of endothelial cells and that reduction of alpha globin genes and hence alpha globin lead to less destruction of nitric oxide (NO) and improved vasodilation. However, deoxy alpha globin can act as a nitrite reductase and can produce NO from nitrite, leading to vasodilation. Hence under conditions of hypoxia, reduced levels of endothelial alpha globin would lead to decrease ability to vasodilate. So, the effects of reduced alpha globin on NO are opposite in the presence of oxygen and in the absence of oxygen. In this paper, the authors address the role of alpha globin using a unique conditional knockdown of the Hba1 gene and endothelial specific Hba1 gene. Overall, the conclusions from this extensive and very elegant series of experiments are very convincing. The findings have very significant implications, especially for humans with alpha thalassemia major (4 gene deletion) who now survive with in utero transfusion.

However, there are a number of questions regarding the work that need to be addressed, and some suggestions that might help an already well-written paper.

First, it would make the paper clearer and easier to understand if a few points were made clearly at the beginning to set the stage as you read the paper. Some of these only came out to me after a couple of readings of the paper. A) the divergent effects of low α -globin in the presence of oxygen, and in the absence of oxygen B) the relation of your knock-downs/outs to clinical α -thalassemia. Your models are equivalent to a trans-two α -gene deletion ($\alpha 1$ - $\alpha 2/\alpha 1$ - $\alpha 2$) which is α -thal trait and has very slightly lower hemoglobin, and very low MCV and usually relatively high RBC. A missing single α -gene has no hematological findings, in humans at least. These findings are only present with there are two α -genes missing, CIS or TRANS. While you don't say it incorrectly, this is only implied by the Hba1-/- superscripts in your abbreviations. So, making this point in the intro would help a clinician at least focus on the superscripts. C) some mention of the differences in α -globin output from $\alpha 1$ and $\alpha 2$ would help.

You appear to have two CRE knockouts: one that is global, and one that is EC specific .. this did not become clear to me until I saw the differences in Supp fig 3 and supp fig 4 ... Perhaps you should use GBL Hba1 Δ/Δ and EC Hba1 Δ/Δ in your discussion and figures to emphasize this. The information is in the paper in line 304, but it is not at all clear, or at least took me a while to get it (maybe I am dense!). Seems a sentence like "thus we have two conditional knock downs, global Hba1 Δ/Δ which affects all HbA1 in erythroid cells and endothelium, and EC Hba1 Δ/Δ which is only in endothelium". I focused on the Δ/Δ and was thinking EC only for a long time.

Fig 2 (b) Hba1 Δ/Δ is clearly different .. but what about α -globin from Hba2 in (d)? I don't understand why you would not see some α -globin if you only affect Hba1 and Hba2++ is still there. You do use the word "reduced" rather than absent α -globin in the text, so I presume it is there but below the limit of visibility by immunofluorescence?

Along those same lines, global Hba1 Δ/Δ should also have reduced EC alpha globin. is this the case? it looks like it in subsequent functional studies you did.

You showed that your construct for deletion of Hba1 Δ/Δ and HbA1-- was specific, but is the same true for the Hba1-36-39? I am surprised that the homozygous state is lethal if HbA2 is still there and not affected and thus wonder if it is affected by the -(36-39) also?

Line 304 your global cre knockout will produce the equivalent of a trans-two-gene alpha trait .. which in humans does not cause significant anemia but low MVC and increased RBC. Why are your RBC lower, I wonder? Again, this may be a mouse vs human difference.

How did you measure the Hb/hct/rbc? there is a problem if you used an automated counter rather than a "spun" hct. automated cell counters measure the MCV and calculate the Hct based on the RBC. two gene deletion alpha thal has a low MCV and a high RBC relative to the degree of anemia. $MCV = (Hct \times 10) / rbc$. I don't think this changes your conclusions, but may affect differences in Hct measures between WT and -- .

Line 381 and figure 4 c: Perhaps I don't understand, but why is the distance to exhaustion for the Hba1fl/fl greater than the Hba1wt/wt ? both have the equivalent amount of α -globin, I think. These differences do not seem to be there for other comparable parameters in fig 4, or in Fig 5 or fig 7. And Supplemental 9 does show a difference between Hba1+/+ and Hba1-/- consistent the idea the presence of alpha globin in fl/fl and wt/wt should have the same distance to exhaustion.

My vote would be to replace figure 6b with supplemental 11. The inclusion of the Hba1+/+ and -/- really puts the whole terrific story in one figure, albeit a bit busier. Up to you.

The discussion is very nice, but there is a fair amount of repetition of the data. It might be about 10-15% shorter.

Minor issues:

Lone 97 should be "confirmed a role"

line 115 do you mean only when AG is deleted from endothelium?

line 191 should be Alexa Fluor ?

Line 313 HbA1 knockdown is specific for HbA1 .. not Hba2 ..

line 352 you state that there is no difference in Hb/Hct or red count in the Hba36-39 trait .. this may not be significant statistically, but it looks close .. and the RBC is lower in the 36-39 trait. I dont know what to expect in the mouse, but the RBC is usually higher than normal in thal-trait than in WT humans. can you explain the difference here?

Line 257 3 0mM KCL ???

line 305 again why would you expect Hba1 to affect myeloid lineage? as opposed to erythroid? this is confusing other than erythroid cells do come from a common myeloid progenitor i guess. You refer to "blood counts" but you are referring to RBC .. again, this is confusing / distracting to me . why not say "RBC counts". Blood counts to me means WBC, RBC, plts and myeloid means white cells.

line 332 the labels in fig D are too small to read.

Line 341 fig 5C is too dark .. cant see it well

Line 387 you looked at capillary density .. can you look at capillary diameter/venous diameter or tortuosity? see PMID 19915173 (not by or related to this reviewer). I bet what they are seeing in muscle biopsy from exercised humans with thal trait is related to your findings here.

Reviewer #4 (Remarks to the Author):

This is an interesting manuscript describing a series of experiments using some elegant transgenic mouse models proposing a novel paradigm where endothelial alpha hemoglobin acts as a nitrite reductase and thus plays an important role in vascular physiology, particularly in relation to hypoxia. The novel concept proposed is that the metabolic context determines the function of alpha hemoglobin i.e. in an oxygenated environment it is one of controlling NO generation via interaction with eNOS and that in a hypoxic environment, where NO levels fall, it functions as a nitrite reductase to deliver much-needed NO.

I have some comments:

1. Generally, whilst the manuscript describes a series of experiments that build a story and that together that story is understandable I have some uncertainties regarding experimental design and analysis that reduce my confidence in the likelihood of the findings being correct. In particular, I was unable to understand the rationale behind experimental design or n values. On several occasions, outcome measures in one genotype involve over 20 independent n's but for the very same outcome measure in another genotype, it is a handful. The problem with this approach is that it can lead to a potential bias in the outcome. There are also a few occasions where quite significant decisions are made seemingly on the basis of an n=1. It is possible that there is an entirely rational explanation for this but without this being articulated in the manuscript the reader is left wondering.
2. Figure 1 shows some lovely images of the two vessel types, the aorta and the thoracodorsal, with respect to the expression of the alpha and beta hemoglobin, and the three other globins. Whilst these images are very clear there is no associated positive control data? How good are these antibodies? Are they selective-there is no detail regarding the antibodies used for all of the globins except for the alpha form in the methods?
3. There are data from clinical tissues but I could find no explanation of from whom these samples came. Were these samples from diseased patients? What are the demographics, was ethical permission in place? Is this just 1 human sample? How can one be confident that this observation is correct if n=1?
4. In the transgenic mice the impact of deletion is clearly shown in the blood vessel wall. However, important controls are missing? What happens to erythrocytic expression in the EC knockout? The authors show RBC parameters unchanged but do not show alpha globin expression?
5. For the Hba136-39 mice, the authors explain that the deletion is embryonic lethal and that there are extreme levels of nitrotyrosine (in what tissues?) that potentially explain this, which did not occur in the heterozygotes. This observation underlies why for all experiments the

heterozygotes were used. The authors go on to suggest that a combination of oxidative and nitrosative stress likely is key to a lack of survival. Whilst this is an interesting proposal it appeared to me that this conclusion was based on an $n=1$?

6. The authors show functional confirmation of EC deletion in various ways including through measurement of NO. For these experiments, vessels were taken and treated with CCh. My question here is why did the WT show absolutely no DAF-response to CCh-this is very unusual-there should have been a response; at this concentration CCh causes a maximal relaxation of these vessels and a significant component of this is mediated by NO. Also, if the deletion raises NO levels then arterial tone should be lower and BP lower too? In addition, it would have been helpful if these measurements had been conducted in the same artery types as above instead of mesenteric arteries, for continuity

7. The statement below is not clear to me. "Relative changes in DAF-FM fluorescence were obtained by dividing the fluorescence in the treatment group by that in the control group. Each field of view was considered as $n=1$, and several fields of view from at least three arteries from at least three mice from each group were included in the final analysis." Does this mean that for statistical analysis that for each mouse 15 values were used to represent $n=15$ rather than $n=1$, and thus in each group there was an $n=45$? Supplemental figure 6 shows only an $n=3$. Which values were used for the statistics, $n=45$ or the $n=3$?

8. The binding affinity experiments assessing binding of the oxygenase domain of eNOS to the WT or 36-39 deletion peptide show a peak that is the same for both genotypes? Again, I think this is a single experiment in triplicate and insufficient data to provide confidence in the finding.

9. The levels of nitrite and nitrate are shown to be different between the genotypes and in response to chemical hypoxia. Do these differences persist if the levels are normalised to sample size or protein concentration?

10. What was the response to NS309 in the different genotypes? When vessels were treated with L-NAME did this change basal tone-since in the Hba136-39 the authors speculate that eNOS activity increases and NO rises. If this is the case one would suspect that there would have been a very large preconstruction to L-NAME treatment?

11. The baseline blood pressures of the different genotypes appear to be different? Perhaps the n number needs to be higher? The issue with this is that if baseline blood pressure is different, then the response to a vasodilator stimulus will vary. A reduced baseline blood pressure will lead to a reduced blood pressure-lowering response. A way to check this would be to see if baseline blood pressure and the response to hypoxia is correlated?

12. Whilst I agree that the data perhaps suggest that the globin acts as a nitrite reductase I think the authors need to demonstrate this. Does purified protein when incubated with nitrite make NO? This would provide clear evidence of the activity. The biological importance of this then could be confirmed using tissues from the transgenics and incubating with nitrite to measure NO generation?

13. What happens to nitrite-induced relaxation in blood vessels collected from these animals in hypoxic conditions?

Response to Reviewers for Nature Communications #NCOMMS-21-08160

Reviewer 1

This is an interesting and potentially important work. I have some points that for the authors to consider.

Is there an overreliance on dithionite in these studies? It is a strong reductant that may change the oxidation state of many biomolecules (and not simply reduce molecular oxygen), including those that may be important for control of vasotone and blood pressure. Even the product formed during dithionite oxidation may, induce oxidation of targets to have an impact on vasodilation. Therefore, can some experiments with authentic hypoxia be performed? Does dithionite in a test tube directly react with nitrite, or in cells is it possible there is some catalysis of this direct reaction?

The original manuscript heavily relied on Na₂S₂O₄ for induction of hypoxia, as it was a reproducible, robust method for hypoxia in many experimental conditions. In response to this comment, we have included experiments with hypoxic solutions produced by bubbling N₂ gas through KREBS buffer to remove oxygen. These experiments can be found in Figures 6 and 7. Thank you for this comment, as it adds to the rigor of the experiments by confirming the results are not artefactual for Na₂S₂O₄ incubation.

It has been reported that Na₂S₂O₄ does not reduce nitrite in Salhany, J.M Biochemistry (2008) PMID: 18465875, so we have referred to this article in the text to justify the use of chemical hypoxia for assaying nitrite reduction on line 320.

Figure 4C – why is the Hba fl/fl so different than the Hba WT/WT.

While we don't have a complete picture of the differences in strain background to complement the differences in running capacity, it has been shown that strain differences can affect exercise capacity (Avila, Kim, Masset Front. Physiol. 2017, PMID 29249981). We have updated the methods to include the description that the Hba1^{fl/fl} mice were backcrossed to a C57Bl/6 background, while the Hba1^{Δ36-39} and their littermate controls were maintained on a mixed background through inbreeding. As such, we have only made comparisons between littermates for these experiments, as those are the proper controls for the experimental groups.

Figure 4D – how can lactate decrease upon exercise in some animals. It seems the EC Hba null shows this loss lactate phenotype more prevalently? Is this a real biological event and if so, what might it mean?

Thank you for this question. We don't know why the Hba null shows loss of lactate more prevalently, however we note that it's not statistically significant between the two groups. The blood that was collected was a mix of arterial and venous (mouse tail) and arterial lactate resolves quickly after cessation of exercise which may add variability to the measurements. We have added a comment to this effect in the manuscript.

Figure 4e – capillary density is double in the Hba WT/delta36-39 mice. What does this mean?

This is an interesting subpoint, but one that we do not have solid data to incorporate. It is known that NO can support angiogenesis in muscle (PMID: 11994253 and 11994243). While the difference in capillary density in the muscle is not significant (thus we have not included further analysis) it could be that higher basal NO production is responsible for greater capillary density in the muscle.

Figure 5E – “NOx” is mentioned – does that include low molecular weight and protein nitrosothiols?

NOx in this context is only the summation of nitrite and nitrate; this point has been clarified in the discussion about (new) Figure 6 and its caption.

Why is the dithionite experimental groups in Figure 5 not also shown under aerobic conditions?

We have updated Figure 5 (now, Figure 6) with new experimental genotypes and added vehicle treatment controls for Na₂S₂O₄ treatment. Additionally, we have demonstrated that dithionite treatment has the same effect as hypoxia induced by treatment with buffer deoxygenated by bubbling with N₂ gas (in WT vessels).

In the updated Figure we also compare the effects of Na₂S₂O₄ with vehicle (H₂O) which represents the normoxic control group. Compared to the normoxic group, nitrite was consumed in vessels rendered hypoxic via Na₂S₂O₄ or N₂ gas treatment. This is especially evident in the comparison of Hba1WT/Δ36-39 +Na₂S₂O₄ and +H₂O conditions.

Figure 6B – do the profiles of vasodilation look the same if normalization is not used and the force data are plotted? It seems from some other data presented in previous Figures that the two transgenics lines have alterations to their vasculature and I am wondering if there are differences in the dilatory or constrictor responses of these vessels that is not apparent due to normalization.

This is an interesting point. The data are generated with a pressure myography system, rather than wire myography, so the %constriction and %relaxation are calculated from digital calipers and a videomicroscopy setup rather than force production on wires. This allows each vessel to behave as it would without alteration of endothelial function by wire pressure on the vessel wall, and thus we believe is a more precise measure of arterial constriction and relaxation. Second, each vessel is normalized to its maximal constriction and dilation assessed by phenylephrine precontraction and Ca²⁺-free buffer for maximal relaxation. This allows interrogation of the function of the nitrite reduction system in affecting dilation to each vessel, rather than arbitrary luminal diameter

measurements that might be affected by slight anatomical changes with vessel size or mouse weight.

To directly address this point, we have added a supplemental figure detailing observations of basal tone of the vessels (supplemental figure 13). Across two models of resistance arteries, the thoracodorsal artery and the 3rd order mesenteric, we do not observe a difference in tone for any genotype.

As mentioned above, it would be good if the data with authentic hypoxia (not chemical hypoxia) can be provided – for example in Figure 6B.

Data have been generated using buffer deoxygenated with N₂ gas for (now) figures 6 and 7, as discussed above. Thank you for this salient point.

Figure 6h utilizes TEMOL – often described as an SOD mimetic. If this is the case, one would expect the production of hydrogen peroxide (a vasoactive molecule) at a higher rate when it is added. What do the authors think about this?

This is an interesting point. TEMPOL may induce relaxation by either increasing NO-bioavailability by preventing the reaction between NO and superoxide, or by promoting the formation of H₂O₂. Conversely, other studies have shown vasoconstrictive effects of TEMPOL skeletal muscle arterioles and in fact demonstrated a biphasic effect with TEMPOL concentration (as described in Cseko, Bagi, and Koller, J Appl Physiol, 2004). The salient observation in our experiments is that TEMPOL had no effect suggesting that superoxide and/ or H₂O₂ are not playing a key role. This was important to exclude as hypoxia may alter superoxide or H₂O₂ metabolism. Data within the same figure point to NO as the major vasoactive molecule (Na₂S₂O₄ + PTIO, N₂ + PTIO, in which PTIO is an NO scavenger) in this experimental setup. Our data show that NO generation, from a non-nitric oxide synthase source (L-NAME inhibition), correlates with alpha globin expression, thus pointing strongly to nitrite reduction mechanisms.

I think it is very important that the in vivo blood pressure measurements are replicated a lot more, having observed treatment effects with low samples sizes. The diurnal data would be of interest, as would the diastolic and mean arterial values. How does heart rate and cardiac output change between genotypes and with hypoxia? For example, are there changes to these parameters that complicate the interpretation of the systolic blood pressure changes reported?

Thank you for these comments. We have increased the number of experimental animals in each of the genotypes from 3-4 to 6-7 animals, and added the global Hba1^{-/-} model to these experiments. Additionally, we have displayed the data in an individualized manner to demonstrate the blood pressure change for individuals under normoxia and hypoxia. Further, we have included the heart rate changes of the individual animals in Figure 8e. This new data shows that compensatory increases in heart rate were seen only in the animals where blood pressure decreased in response to hypoxia. The Hba1^{-/-} and EC Hba1 Δ/Δ groups did not have changed heart rate

because there was no reflex to a decreased blood pressure level. This analysis has been added to the results section and discussion of Figure 8. Last, it should be noted that these blood pressure measurements are an acute snapshot in response to the hypoxia. Circadian rhythms with a slight increase in blood pressure at night and lower during the day were normal and noted in text.

Reviewer 2

Keller and colleagues report studies of the dual functions of endothelial cell (EC) Hba1 related to nitrite reduction and eNOS-derived NO scavenging. They employ mice with EC-specific deletion of Hba1 and mice with mutant Hba1 incapable of inhibitory association with eNOS. The studies were carefully carried out using standard methods, and they demonstrate that EC Hba1 is required for hypoxia-related vasodilation in isolated arteries, and for the hypoxia-related decline in systolic BP. The reported observations regarding EC Hba1 and nitrite consumption by hypoxic isolated vessels are also convincing, but the data would be strengthened if it included results for vessels in all genotype groups studied under normoxic conditions. The changes in exercise tolerance observed in mice with EC Hba1 deletion are modest. The primary limitation of the work is that firm conclusions cannot be drawn about EC Hba1 NO scavenging because fetal lethality required that the Hba1 mutant mouse studies be done in heterozygous mice. Other concerns are raised about data presentation and interpretation.

SPECIFIC COMMENTS

Abstract

1. It should be more clearly mentioned in the abstract that the studies of the mutant mice were done in heterozygotes.

This has been made clearer through additional mention of this in the abstract.

Methods

2. Why were the studies of hypoxia-related vasodilation limited to the use of chemical hypoxia? This should be stated.

Na₂S₂O₄ is a commonly used O₂ scavenger. In addition to Na₂S₂O₄, in the revised manuscript we show new experimental data using buffer bubbled with N₂ gas where to induce hypoxia. The results from Na₂S₂O₄ and N₂-treated buffer are highly concordant; see Figures 6 and 7 in the revised manuscript.

Results

3. What was the sample source for the PCR amplification studies shown in Fig. 2B?

The PCR amplification was from genomic DNA collected from diaphragm tissue, where endothelial cells make up a prominent cell population. Non-deletion bands were likely due to cell populations that do not express the *Cdh5-CreERT2* driver.

4. Fig. 2C- the location of EC Hba1 at myoendothelial junctions is not convincing (and there are no related breaks in the internal elastic lamina). This feature needs to be better visualized in order to make this claim.

Thank you for this comment; we agree that transverse sections of the thoracodorsal artery were insufficient to claim localization of alpha globin in a small domain of endothelium. We have included two additional data representations to demonstrate the expression of alpha globin at the myoendothelial junctions (MEJs). First, *en face* confocal images are added to (now) Figure 3c-e to show alpha globin expression in the holes of the internal elastic lamina. Additionally, we have analyzed these sections from XZ and YZ planes using confocal microscopy to demonstrate colocalization of alpha globin signal in the plane of the IEL, and where lamina signal is absent (Supplemental Figure 5, new data).

5. Suppl. Fig. 5C- what tissue was imaged? This should be mentioned at least at a minimum in the legend.

(Now Supplemental Figure 8) Thank you for the opportunity to clarify this experiment. The entire embryo section was imaged for quantification, but the representative images are of the embryonic heart and trunk vasculature. This has been added to the figure caption.

6. Suppl. Fig. 6B- treatment groups should be labeled in the graph.

(Now Supplemental Figure 11) The figure caption has been updated to reflect the treatment conditions in the graph, and the graph text has been increased in size to be more legible.

7. Fig. 4C- why is there a difference in distance to exhaustion in the two control groups (Hba1 f/f versus Hba1 wt/wt)?

While we don't have a complete picture of the differences in strain background to complement the differences in running capacity, it has been shown that strain differences can affect exercise capacity (Avila, Kim, Masset Front. Physiol. 2017, PMID 29249981). We have updated the methods to include the description that the Hba1fl/fl mice were backcrossed to a C57Bl/6 background, while the Hba1 Δ 36-39 and their littermate controls were maintained on a mixed background through inbreeding. As such, we have only made comparisons between littermates for these experiments, as those are the proper controls for the experimental groups.

8. Line 388-390- It is not possible to conclude that preventing Hba1-eNOS association does not affect exercise capacity because the loss-of-function is not complete.

We agree that this was overstating the conclusions from the data. This statement has been removed from the revised manuscript.

9. Fig. 5- data are needed for H2O-only treated arteries in each genotype group so changes in nitrite consumption with chemical hypoxia can be evaluated.

Thank you for this critical point. In the revised figure (now Figure 6), we have included H₂O treatment as the vehicle control for Na₂S₂O₄ treatment in the experimental genotypes. This ablates the ability of any group to consume nitrite in this experiment, and is a good visualization for the requirement of hypoxia in the system.

10. Fig. 6- the authors need to provide evidence of effective antagonism of eNOS-dependent vasodilation with L-NAME in the specific model system employed.

In Figure 6e (now as Figure 7e), we have included L-NAME treatment to rule out NOS-dependent NO production. Importantly, in the revised manuscript, we have extended this experiment to both hypoxia induced by Na₂S₂O₄ and by deoxygenated KREBS buffer (by bubbling the buffer with N₂ gas). In both experiments, L-NAME does not inhibit the relaxation induced by hypoxia that depends on endothelial alpha globin expression (global *Hba1*^{-/-} and EC *Hba1*^{ΔΔ} genotypes) and NO generation (via PTIO treatment).

11. Fig. 6H, and line 433- instead of “baseline dilation...”, should the sentence read “hypoxic vasodilation...”?

Thank you for this clarifying edit, it has been incorporated into the revised manuscript.

Discussion

12. Lines 478-479- Although limitations of the work are appropriately mentioned later in the Discussion, here “we are able to specifically assay a nitrite reductase role for endothelial Hba1” is an overstatement.

With the addition of data in the revised manuscript Figure 2, we can confirm nitrite reduction activity by endothelial alpha globin. This confirmatory *in vitro* data, along with the functional data in figures 6, 7, and 8, are hopefully solid enough in these conclusions to merit inclusion of this statement.

13. Line 486- differences in running distance between Hba1 fl/fl controls and Hba1 wt/wt controls should be mentioned here.

This statement has been updated in the discussion of the revised manuscript, at line 571.

Reviewer 3

Keller and colleagues have undertaken an extensive and well thought out series of experiments to explore the role of “Endothelial alpha globin as a nitrite reductase”. This work is based upon the initial observation by Straub and Isakson in 2012 that alpha globin is expressed in the myoendothelial junction of endothelial cells and that reduction of alpha globin genes and hence alpha globin lead to less destruction of nitric oxide (NO) and improved vasodilation. However, deoxy alpha globin can act as a nitrite reductase and can produce NO from nitrite, leading to vasodilation. Hence under conditions of hypoxia, reduced levels of endothelial alpha globin would lead to decrease ability to vasodilate. So, the effects of reduced alpha globin on NO are opposite in the presence of oxygen and in the absence of oxygen. In this paper, the authors address the role of alpha globin using a unique conditional knockdown of the Hba1 gene and endothelial specific Hba1 gene. Overall, the conclusions from this extensive and very elegant series of experiments are very convincing. The findings have very significant implications, especially for humans with alpha thalassemia major (4 gene deletion) who now survive with in utero fusion.

However, there are a number of questions regarding the work that need to be addressed, and some suggestions that might help an already well-written paper.

First, it would make the paper clearer and easier to understand if a few points were made clearly at the beginning to set the stage as you read the paper. Some of these only came out to me after a couple of readings of the paper. A) the divergent effects of low α -globin in the presence of oxygen, and in the absence of oxygen B) the relation of your knock-downs/outs to clinical α -thalassemia. Your models are equivalent to a trans-two α -gene deletion ($\alpha 1$ - $\alpha 2/\alpha 1$ - $\alpha 2$) which is α -thal trait and has very slightly lower hemoglobin, and very low MCV and usually relatively high RBC. A missing single α -gene has no hematological findings, in humans at least. These findings are only present with there are two α -genes missing, CIS or TRANS. While you don't say it incorrectly, this is only implied by the Hba1-/- superscripts in your abbreviations. So, making this point in the intro would help a clinician at least focus on the superscripts. C) some mention of the differences in α -globin output from $\alpha 1$ and $\alpha 2$ would help.

Thank you for the interesting clinical perspective to add to the introduction. In response to (A), a sentence has been added to the introduction at the top of page 2, line 106-110. In response to (B), we have added this insight into the results section of Figure 3. In response to (C), we do not know specifically how each of Hba1 and Hba2 contribute to endothelial or myeloid gene expression. However, the lack of endothelial alpha globin immunohistochemistry in each of the global neomycin induced deletion (Hba1-/-) and endothelial-specific (EC Hba1 Δ/Δ) knockouts points to a major contribution of Hba1 to endothelial alpha globin protein. This, combined with the lack of lethality from the global Hba1 Δ/Δ (Hba1^{fl/fl}; Sox2-Cre) confirms that we are targeting Hba1 with the loxP sites and that is sufficient to reduce endothelial alpha globin.

You appear to have two CRE knockouts: one that is global, and one that is EC specific .. this did not become clear to me until I saw the differences in Supp fig 3 and supp fig 4

... Perhaps you should use GBL Hba1 Δ/Δ and EC Hba1 Δ/Δ in your discussion and figures to emphasize this. The information is in the paper in line 304, but it is not at all clear, or at least took me a while to get it (maybe I am dense!). Seems a sentence like “thus we have two conditional knock downs, global Hba1 Δ/Δ which affects all HbA1 in erythroid cells and endothelium, and EC Hba1 Δ/Δ which is only in endothelium”. I focused on the Δ/Δ and was thinking EC only for a long time.

We have added the “global” appendage to the discussion of results from the Hba1 Δ/Δ mouse, as requested. This is mostly in the section about (now) figure 3, as this model is only used for comparison of blood parameters from EC and neomycin-induced global knockout models.

Fig 2 (b) Hba1 Δ/Δ is clearly different .. but what about α -globin from Hba2 in (d)? I don't understand why you would not see some α -globin if you only affect Hba1 and Hba2 $^{++}$ is still there. You do use the word “reduced” rather than absent α -globin in the text, so I presume it is there but below the limit of visibility by immunofluorescence?

As above, we demonstrate that our conditional allele is only targeting Hba1 (through lethality studies with the Sox2-Cre driver), and thus Cdh5-CreERT2-mediated deletion is sufficient to reduce alpha globin below the detectable limit. While we would like to assume that all endothelial alpha globin is gone with Cre-mediated deletion, it is possible that some cells escape deletion and express alpha globin. However, we do not see any endothelial alpha globin from either the neomycin-knockout Hba1 $^{-/-}$ or our EC Hba1 Δ/Δ (Supplemental Figure 4 and now Figure 3d) in the endothelium, so we do not presume that Hba2 is contributing significantly to endothelial alpha globin. Our use of “reduced” simply points to the (relatively low, but still present) inefficiency in knockout. It is possible in other organisms Hba2 may contribute. We are still deciphering endothelial alpha globin expression/biophysics/function, but are very excited by the findings thus far.

Along those same lines, global Hba1 Δ/Δ should also have reduced EC alpha globin. is this the case? it looks like it in subsequent functional studies you did.

Thank you for this observation. The global Hba1 Δ/Δ animals do have reduced endothelial alpha globin, but also suffer from alpha thalassemia with hemolytic anemia which confounds results used in exercise studies so we have not included this data.

You showed that your construct for deletion of Hba1 Δ/Δ and HbA1 $^{--}$ was specific, but is the same true for the Hba1-36-39? I am surprised that the homozygous state is lethal if HbA2 is still there and not affected and thus wonder if it is affected by the $-(36-39)$ also?

It is possible that the Hba2 gene is also affected by this CRISPR-based mutation. However, erythrocytes numbers and characteristics appear unaffected by this mutation (supplemental figure 9). We do not see a significant contribution of Hba2 to endothelial

alpha globin expression, so it is unclear whether this mutation is affecting both alleles. However, this is outside the scope

Line 304 your global cre knockout will produce the equivalent of a trans-two-gene alpha trait .. which in humans does not cause significant anemia but low MVC and increased RBC. Why are your RBC lower, I wonder? Again, this may be a mouse vs human difference.

In our global Hba1 Δ/Δ mutant, we observe a significant anemia and low MVC, but not significantly reduced RBC counts (supplemental figure 7). We assume this is due to loss of the trans-two gene deletion, but the specific differences might be due to mouse vs. human control of erythrocyte characteristics.

How did you measure the Hb/hct/rbc? there is a problem if you used an automated counter rather than a "spun" hct. automated cell counters measure the MCV and calculate the Hct based on the RBC. two gene deletion alpha thal has a low MCV and a high RBC relative to the degree of anemia. $MCV = (Hct \times 10) / rbc$. I don't think this changes your conclusions, but may affect differences in Hct measures between WT and -/- .

We agree with the reviewer on this comment and decided to evaluate from Hba1 $^{-/-}$ and control mice how the method may influence the values. We observed from both genotypes that the automated measurement slightly underestimated the Hct compare to spun Hct. We decided to maintain the data presented as the error of sharing automated measurement of Hct was small and did not affect overall our conclusions. See the figure below:

Automated and spun blood counts are highly concordant for the Hba1 $^{-/-}$ and Hba1 $^{+/+}$ genotypes.

Line 381 and figure 4 c: Perhaps I don't understand, but why is the distance to exhaustion for the Hba1fl/fl greater than the Hba1wt/wt ? both have the equivalent amount of α -globin, I think. These differences do not seem to be there for other comparable parameters in fig 4, or in Fig 5 or fig 7. And Supplemental 9 does show a difference between Hba1+/+ and Hba1-/- consistent the idea the presence of alpha globin in fl/fl and wt/wt should have the same distance to exhaustion.

While we don't have a complete picture of the differences in strain background to complement the differences in running capacity, it has been shown that strain differences can affect exercise capacity (Avila, Kim, Masset Front. Physiol. 2017, PMID 29249981). We have updated the methods to include the description that the Hba1fl/fl mice were backcrossed to a C57Bl/6 background, while the Hba1 Δ 36-39 and their littermate controls were maintained on a mixed background through inbreeding. As such, we have only made comparisons between littermates for these experiments, as those are the proper controls for the experimental groups.

My vote would be to replace figure 6b with supplemental 11. The inclusion of the Hba1+/+ and -/- really puts the whole terrific story in one figure, albeit a bit busier. Up to you.

We have included the use of the neomycin Hba1-/- model in the rest of the panels in (now) figure 7, and have replaced the panel in B with the full data set. Thank you for the comment that spurred this change.

The discussion is very nice, but there is a fair amount of repetition of the data. It might be about 10-15% shorter.

Critical revisions have occurred, thank you.

Minor issues:

Line 97 should be "confirmed a role"

Fixed.

line 115 do you mean only when AG is deleted from endothelium?

Thank you for the clarifying edit, the line now reads when "alpha globin is deleted from the endothelium".

line 191 should be Alexa Fluor ?

This has been clarified in the methods. Thank you

Line 313 HbA1 knockdown is specific for HbA1 .. not Hba2 ..

This typo has been fixed. Thank you for your keen eye on this and other typos.

line 352 you state that there is no difference in Hb/Hct or red count in the Hba36-39 trait .. this may not be significant statistically, but it looks close .. and the RBC is lower in the 36-39 trait. I dont know what to expect in the mouse, but the RBC is usually higher than normal in thal-trait than in WT humans. can you explain the difference here?

We agree the model does not fully recapitulate an alpha thalassemic trait, even with observed trends toward significant differences in Hb, Hct, and RBC. The differences are not statistically significant, however, so we feel comfortable including this data as is.

It is possible that although the Δ 36-39 protein is expressed and does not immediately cause precipitation in the erythrocyte (Supplemental figure 9d-f), it causes the erythrocyte to become slightly more fragile and thus reduces numbers, although not statistically different.

Line 257 3 0mM KCL ???

This typo has been fixed.

line 305 again why would you expect Hba1 to affect myeloid lineage? as opposed to erythroid? this is confusing other than erythroid cells do come from a common myeloid progenitor i guess. You refer to "blood counts" but you are referring to RBC .. again, this is confusing / distracting to me . why not say "RBC counts". Blood counts to me means WBC, RBC, plts and myeloid means white cells.

This instance of myeloid has been changed to "hematopoietic"; thank you for the clarifying comment. We have also changed "blood count" to "RBC count" in the same section.

line 332 the labels in fig D are too small to read.

We have enlarged in his resubmission.

Line 341 fig 5C is too dark .. cant see it well

We have updated figures for clarity.

Line 387 you looked at capillary density .. can you look at capillary diameter/venous diameter or tortuosity? see PMID 19915173 (not by or related to this reviewer). I bet what they are seeing in muscle biopsy from exercised humans with thal trait is related to your findings here.

This is an interesting point, thank you for the future direction. We have not analyzed tortuosity in this manuscript, but will in future publications to examine the role of endothelial alpha globin in controlling NO signaling during arterial development. It is

known that NO can support angiogenesis, so this is an interesting future direction for these models.

Reviewer 4

This is an interesting manuscript describing a series of experiments using some elegant transgenic mouse models proposing a novel paradigm where endothelial alpha hemoglobin acts as a nitrite reductase and thus plays an important role in vascular physiology, particularly in relation to hypoxia. The novel concept proposed is that the metabolic context determines the function of alpha hemoglobin i.e. in an oxygenated environment it is one of controlling NO generation via interaction with eNOS and that in a hypoxic environment, where NO levels fall, it functions as a nitrite reductase to deliver much-needed NO.

I have some comments:

1. Generally, whilst the manuscript describes a series of experiments that build a story and that together that story is understandable I have some uncertainties regarding experimental design and analysis that reduce my confidence in the likelihood of the findings being correct. In particular, I was unable to understand the rationale behind experimental design or n values. On several occasions, outcome measures in one genotype involve over 20 independent n's but for the very same outcome measure in another genotype, it is a handful. The problem with this approach is that it can lead to a potential bias in the outcome. There are also a few occasions where quite significant decisions are made seemingly on the basis of an n=1. It is possible that there is an entirely rational explanation for this but without this being articulated in the manuscript the reader is left wondering.

In the revised manuscript, we have added repeated measures of multiple experiments, including in figures 6 and 7. Thank you for the opportunity to strengthen these experiments.

2. Figure 1 shows some lovely images of the two vessel types, the aorta and the thoracodorsal, with respect to the expression of the alpha and beta hemoglobin, and the three other globins. Whilst these images are very clear there is no associated positive control data? How good are these antibodies? Are they selective-there is no detail regarding the antibodies used for all of the globins except for the alpha form in the methods?

We have updated the methods section to reflect the antibodies used. Thank you for the opportunity to clarify this section of the methods.

3. There are data from clinical tissues but I could find no explanation of from whom these samples came. Were these samples from diseased patients? What are the demographics, was ethical permission in place? Is this just 1 human sample? How can one be confident that this observation is correct if n=1?

Human samples were not extensively tested (e.g., at the functional level), and were used simply to demonstrate murine similarity to human tissue. These particular samples were obtained from normotensive adipose tissue biopsies as part of another study in the

lab, and analysis specific to this manuscript was performed. The representative images are indicative of at least 3 patients in all cases. This information has been added to the figure caption. Additional information was placed in the text.

4. In the transgenic mice the impact of deletion is clearly shown in the blood vessel wall. However, important controls are missing? What happens to erythrocytic expression in the EC knockout? The authors show RBC parameters unchanged but do not show alpha globin expression?

Thank you for this comment. In the revised manuscript, we have shown via western blot that hemoglobin content in blood from the EC Hba1 Δ/Δ is unchanged after deletion (Supplemental Figure 6a). This data, combined with the other characterization of the blood parameters and a wealth of studies using the Cdh5-CreERT2 from ours and other labs, demonstrate the endothelial specificity of the deletion.

5. For the Hba136-39 mice, the authors explain that the deletion is embryonic lethal and that there are extreme levels of nitrotyrosine (in what tissues?) that potentially explain this, which did not occur in the heterozygotes. This observation underlies why for all experiments the heterozygotes were used. The authors go on to suggest that a combination of oxidative and nitrosative stress likely is key to a lack of survival. Whilst this is an interesting proposal it appeared to me that this conclusion was based on an n=1?

Thank you for this comment. While the data showing nitrotyrosine staining in the heart and trunk of the E12.5 heterozygous embryo (Supplemental figure 8, with added description of the tissues in the figure caption) is only from n=1, the embryonic lethality was confirmed in supplemental table 5, where from 60 pups genotyped at P21, only 1 could be confirmed as Hba1 Δ 36-39/ Δ 36-39. So, although the cause of lethality is not confirmable with the data included in the manuscript (and indeed, outside the scope of the manuscript as we are not focused on contributions of NO signaling to forming the vasculature in the embryo), we have significant data showing that the homozygous deletion is not viable. That the homozygous mutants did not live was the reason for using heterozygous individuals, as well, not because of increased nitrotyrosine in the embryo sections that we analyzed.

6. The authors show functional confirmation of EC deletion in various ways including through measurement of NO. For these experiments, vessels were taken and treated with CCh. My question here is why did the WT show absolutely no DAF-response to CCh-this is very unusual-there should have been a response; at this concentration CCh causes a maximal relaxation of these vessels and a significant component of this is mediated by NO. Also, if the deletion raises NO levels then arterial tone should be lower and BP lower too? In addition, it would have been helpful if these measurements had been conducted in the same artery types as above instead of mesenteric arteries, for continuity

Thank you for the opportunity to clarify these experiments. First, the DAF experiments (now in supplemental figure 11) were conducted in the Hba1WT/ Δ 36-39 model and thus do not represent a deletion of alpha globin but a partial uncoupling of alpha globin/eNOS binding (heterozygous mutation). Second, it has been established in other manuscripts that NO is not a significant contributor to resistance artery relaxation, even under CCh stimulation, differing from large vessel dilation mechanisms (see: PMID: 30900949; PMID: 32463112) We believe that this is due to the presence of alpha globin in the resistance arteries but not in the large conduit arteries. With the Δ 36-39 heterozygous mutation, we have partially decreased the influence of alpha globin on endothelial NOS production of NO, which is the consequence of cholinergic stimulation in these vessels. Thus, when alpha globin can bind to eNOS, CCh treatment does not increase NO levels that can be detected with DAF fluorescence because alpha globin reacts and oxidizes NO faster than that of the reaction of NO with DAF. However, when some of the pool of endothelial alpha globin cannot interact with eNOS, some NO is available to react with DAF.

Functionally, blood pressure might be different in the Δ 36-39 animals, but because this is a constitutive mutation (rather than an inducible deletion), we cannot account for compensation that might occur before adulthood to normalize blood pressure at appropriate setpoints for tissue perfusion. We note there is a slight reduction in systolic blood pressure, but it is not significant.

The different tissue used does not meaningfully change the results, as the response of the skeletal muscle artery (thoracodorsal) and mesenteric resistance artery are consistent in NO signaling (PMID: 22335567). Additionally, we felt it was important to assay the arteries from the same animals, so using the artery for DAF experiments and vasoreactivity were not possible.

7. The statement below is not clear to me. "Relative changes in DAF-FM fluorescence were obtained by dividing the fluorescence in the treatment group by that in the control group. Each field of view was considered as n=1, and several fields of view from at least three arteries from at least three mice from each group were included in the final analysis." Does this mean that for statistical analysis that for each mouse 15 values were used to represent n=15 rather than n=1, and thus in each group there was an n=45? Supplemental figure 6 shows only an n=3. Which values were used for the statistics, n=45 or the n=3?

We apologize for the typo. Each artery, not each field of view, was considered as n=1. Several fields of view from each artery were averaged to get one averaged fluorescence value for that artery. Thus, n=3 represents data from 3 arteries, a number that was used for statistics. One artery was used per mouse. We have made these changes in the revised manuscript.

8. The binding affinity experiments assessing binding of the oxygenase domain of eNOS to the WT or 36-39 deletion peptide show a peak that is the same for both

genotypes? Again, I think this is a single experiment in triplicate and insufficient data to provide confidence in the finding.

While the data in the manuscript shows a single experiment for clarity, the experiment itself has been replicated across many protein preparations, experimental conditions and treatments (including another published study, Keller et al, Hypertension, 2016 PMID 27802421). The data have been consistent across many replicates, showing a high affinity of the original peptide, and a loss of binding with the $\Delta 36-39$ mutant. A sentence defining the results in the figure as indicative of $n > 5$ experiments has been added to the figure caption. Hopefully this is sufficient to ensure confidence in the result.

9. The levels of nitrite and nitrate are shown to be different between the genotypes and in response to chemical hypoxia. Do these differences persist if the levels are normalized to sample size or protein concentration?

In this experiment, thoracodorsal artery segments were taken at the same site and with consistency in size to demonstrate the effect of chemical hypoxia on NO₂ and NO₃ levels with hypoxic treatment. In the revised figure 6, the total NO₂ and NO₃ are highly concordant across experimental groups and increased N in each group. Additionally, we have added more genetic models and vehicle control experiments to increase confidence in these results.

10. What was the response to NS309 in the different genotypes? When vessels were treated with L-NAME did this change basal tone-since in the Hba136-39 the authors speculate that eNOS activity increases and NO rises. If this is the case one would suspect that there would have been a very large preconstruction to L-NAME treatment?

Thank you for the opportunity to clarify these experiments. The response to 1 μ M NS309 causes a maximum vasodilation and we now have that data in Supplementary Figure 13. L-NAME typically does not cause changes in basal tone of small resistance arteries due to their predominate use of EDH (as opposed to larger arteries that are NO-dependent). L-NAME dependent inhibition is observed only when eNOS is activated. In the Hba1WT/ $\Delta 36-39$ mice, we do not see increased activity of eNOS at baseline (Supplemental Figure 11 without stimulation), only with activation of eNOS after cholinergic stimulation. Thus, in the vasoreactivity experiments in Figure 7, L-NAME does not cause pre-constriction, and does not inhibit the dilation when hypoxia is induced.

11. The baseline blood pressures of the different genotypes appear to be different? Perhaps the n number needs to be higher? The issue with this is that if baseline blood pressure is different, then the response to a vasodilator stimulus will vary. A reduced baseline blood pressure will lead to a reduced blood pressure-lowering response. A way to check this would be to see if baseline blood pressure and the response to hypoxia is correlated?

Thank you for this point concerning blood pressure. In response to this comment, we have increased N for the blood pressure experiments and added the Hba1^{-/-} global knockout genotypes for comparison. Additionally, we have plotted the individualized data in Figure 8d, showing the individual changes of animals when exposed to hypoxia. Even with reduced systolic blood pressure at baseline, the change of the Hba1^{-/-} and EC Hba1 Δ/Δ groups when exposed to hypoxia is near zero and the individuals do not experience severe changes as seen in the littermate controls and Hba1^{WT/ Δ 36-39} animals. We believe that this data is convincing to show that endothelial alpha globin is necessary for the vasodilatory response when the animal is exposed to hypoxia.

12. Whilst I agree that the data perhaps suggest that the globin acts as a nitrite reductase I think the authors need to demonstrate this. Does purified protein when incubated with nitrite make NO? This would provide clear evidence of the activity. The biological importance of this then could be confirmed using tissues from the transgenics and incubating with nitrite to measure NO generation?

Thank you for this comment. In the revised manuscript, we have added data from in vitro experiments that directly address whether alpha globin can reduce nitrite and form NO as a monomer. Indeed, alpha globin can reduce nitrite with greater efficiency than the $\alpha 2\beta 2$ hemoglobin tetramer (Figure 2b). Also, we have demonstrated the ability of lysates from the thoracodorsal artery to form NO when exposed to hypoxia. We believe that these results are sufficient to demonstrate the ability of endothelial alpha globin to produce NO from nitrite.

13. What happens to nitrite-induced relaxation in blood vessels collected from these animals in hypoxic conditions?

Our experiments in Figure 6 demonstrate that intracellular nitrite exists and can be consumed by endothelial cells in a hypoxia and alpha globin-dependent manner. Additionally, nitrite is not added to the buffer in the vasoreactivity experiments in Figure 7, so there is sufficient intracellular nitrite for alpha globin to consume and form vasodilatory NO in this experiment.

REVIEWER COMMENTS

Reviewer #1 (Remarks to the Author):

The authors have fully addressed my comments either by providing rational explanations or by adding in significant additional experiments.

Reviewer #2 (Remarks to the Author):

Keller et al have effectively addressed the vast majority of the concerns raised about the prior version of the manuscript, often by the inclusion of findings from additional experiments. Recognizing that there are limitations in the degree of loss-of-function for endothelial cell (EC) Hba1 inhibitory association with eNOS (as stated by the authors), the collective findings reported demonstrate that EC Hba1 functions as a nitrite reductase which plays a meaningful role in hypoxia-related vasodilation.

SPECIFIC COMMENTS

1. In the legend for Fig. 3b it should be stated that the source of genomic DNA is diaphragm.
2. The confocal localization of alpha globin is quite helpful (Fig. 3c-e, Suppl. Fig. 5).
3. The text description of Fig. 7c (lines 483-485) should be clarified because the phrase "with pharmacologic agents" can be confusing.
4. Fig. 7d: a helpful control would be to compare the effects of WT versus Hba global KO erythrocytes.
5. Fig. 7e-h: control studies are needed to confirm the efficacy of the pharmacologic interventions that lack impact on hypoxic vasodilation. In the same model system it can be evaluated if L-NAME used in the same manner inhibits vasodilation with acetylcholine.
6. Fig. 7i-j: The studies employing N2 nicely confirm the findings with Na₂S₂O₄.

Reviewer #3 (Remarks to the Author):

This is a revised manuscript regarding the role of alpha globin in regulation of vasodilation in resistance vessels.

The authors have addressed all of my concerns and i have no further recommendations or critique.

The physiology uncovered by the authors likely has significant relevance to human vascular disease. A huge percent of the worlds population is missing 1 to 3 alpha globin genes (30% of Africans, 30-40% of some Asian populations). This work is also relevant to humans who survive delivery with 4 missing alpha genes.

Reviewer #5 (Remarks to the Author):

The responses to the concerns expressed by the previous four reviewers have been appropriate and highly beneficial with respect to enhancing all aspects of the manuscript. This manuscript describes what is likely to be paradigm change in understanding how vascular tone and compliance are regulated by several physiological and genetic parameters. With the corrections, additions and edits stimulated by the thoughtful comments of earlier reviewers the manuscript is impressively clear as to methodologies, analysis, background and conclusions. This important contribution warrants publication and the current revised iteration of the manuscript appears to have addressed the several issues that have hindered acceptance. I will reading this manuscript many times in order to fully digest the ramifications with respect to many clinical indications and therapeutic strategies.

Response to Reviewers for Nature Communications #NCOMMS-21-08160

Reviewer 1

The authors have fully addressed my comments either by providing rational explanations or by adding in significant additional experiments.

Thank you for your previous comments which have improved the manuscript.

Reviewer 2

Keller et al have effectively addressed the vast majority of the concerns raised about the prior version of the manuscript, often by the inclusion of findings from additional experiments. Recognizing that there are limitations in the degree of loss-of-function for endothelial cell (EC) Hba1 inhibitory association with eNOS (as stated by the authors), the collective findings reported demonstrate that EC Hba1 functions as a nitrite reductase which plays a meaningful role in hypoxia-related vasodilation.

SPECIFIC COMMENTS

1. In the legend for Fig. 3b it should be stated that the source of genomic DNA is diaphragm.

The highlighted text has been added to the figure 3 caption:

...(b) DNA gel **using genomic DNA extracted from diaphragm** showing recombination of the Hba1 locus with Cre activation . The recombination event produced a band at ~450 bp...

2. The confocal localization of alpha globin is quite helpful (Fig. 3c-e, Suppl. Fig. 5).

Thank you, this was a fun experiment to include.

3. The text description of Fig. 7c (lines 483-485) should be clarified because the phrase "with pharmacologic agents" can be confusing.

Thank you for this comment to help readability of the manuscript. The highlighted text has been updated in lines 483 onward to clarify the sentence:

To determine the source of vasodilatory signals, we tested the vessel dilatory response in the presence of various pharmacologic **agents targeting NO signaling and enzymatic production of NO.**

4. Fig. 7d: a helpful control would be to compare the effects of WT versus Hba global KO erythrocytes.

Adding KO erythrocytes would be interesting, but ultimately answers a different question than we are asking using WT erythrocytes in Fig 7d. Including Hba1 KO erythrocytes

(as Hba1/2 double KO mice are not viable) would better answer the relative contributions of Hba1 and Hba2 in erythrocytes, and even then, is not the right assay for that as β globin has been shown to have nitrite reduction capacity (see papers and reviews by Gladwin and Tejero; and us: <https://doi.org/10.1152/physrev.00037.2020>).

Hba1 Δ/Δ or Hba1 $-/-$ erythrocytes also have Hba2-containing hemoglobin tetramers and are thus not totally lacking hemoglobin, so we presume would act similarly to the WT erythrocytes we used.

The contributions of β 4 globin tetramers to nitrite reduction in erythrocytes is also possible, but those questions do not have a place in this manuscript.

5. Fig. 7e-h: control studies are needed to confirm the efficacy of the pharmacologic interventions that lack impact on hypoxic vasodilation. In the same model system it can be evaluated if L-NAME used in the same manner inhibits vasodilation with acetylcholine.

The effects of L-NAME on ACh-induced vasodilation have been published previously, and thus we did not include it here. Specifically, from our lab, we have used L-NAME to assess vasoreactivity to acetylcholine in the Hba1 $-/-$ model previously: see figure 4H and 4I from Shu, Ruddiman, et al in Circ Research, 2019 (relevant figure panels are included below).

In this experiment, we induced alpha globin expression in the carotid artery by application of PAI-1 to the mural wall through a surgical model. After 7 days, alpha globin was clearly expressed in the endothelium of the carotid artery, where it is normally absent in mice. Using the global Hba1 deletion model from this paper (Hba1 $-/-$ mice – where alpha globin was not induced in the endothelium due to genetic ablation), we were able to determine that L-NAME can significantly inhibit vasodilation to acetylcholine when alpha globin is not present (Panel I, blue line). Thus, we expect a similar phenotype in our hands for the current paper for the global models. We also expect this phenotype to extend to the endothelial-specific knockout of Hba1 used in the current study.

Further, in this manuscript, we have used orthogonal pharmacologic approaches to determine whether NO is the signaling molecule (by PTIO), and whether it is affected by inhibiting NOS enzymes (by L-NAME) or gaseous signaling (by CO). Previous studies have confirmed the importance of eNOS-derived NO for ACh-induced dilation, but this new paradigm for nitrite reduction and endothelial-autonomous hypoxia sensing and vasomotor control is independent of eNOS.

6. Fig. 7i-j: The studies employing N2 nicely confirm the findings with Na2S2O4.

Thank you for previous comments to improve this aspect of the manuscript.

Reviewer 3

This is a revised manuscript regarding the role of alpha globin in regulation of vasodilation in resistance vessels.

The authors have addressed all of my concerns and i have no further recommendations or critique.

The physiology uncovered by the authors likely has significant relevance to human vascular disease. A huge percent of the worlds population is missing 1 to 3 alpha globin genes (30% of Africans, 30-40% of some Asian populations). This work is also relevant to humans who survive delivery with 4 missing alpha genes.

Thank you for your previous comments which have improved the manuscript significantly.

Reviewer 5

The responses to the concerns expressed by the previous four reviewers have been appropriate and highly beneficial with respect to enhancing all aspects of the manuscript. This manuscript describes what is likely to be paradigm change in understanding how vascular tone and compliance are regulated by several physiological and genetic parameters. With the corrections, additions and edits stimulated by the thoughtful comments of earlier reviewers the manuscript is impressively clear as to methodologies, analysis, background and conclusions. This important contribution warrants publication and the current revised iteration of the manuscript appears to have addressed the several issues that have hindered acceptance. I will reading this manuscript many times in order to fully digest the ramifications with respect to many clinical indications and therapeutic strategies.

Thank you for your positive outlook on the future of the manuscript! We believe this is an important study to publish for future vascular biology paradigms.

REVIEWERS' COMMENTS

Reviewer #2 (Remarks to the Author):

The authors have effectively addressed my final set of questions and recommendations. They are to be congratulated on a valuable contribution to our understanding of vasodilation in response to hypoxia.